EMBO
Molecular Medicine

# Impairment of the ER/mitochondria compartment in human cardiomyocytes with PLN p.Arg14del mutation

Friederike Cuello[1,2,†] (ID), Anika E Knaust[1,2,†], Umber Saleem[1,2], Malte Loos[1,2], Janice Raabe[1,2], Diogo Mosqueira[3] (ID), Sandra Laufer[1,2], Michaela Schweizer[4], Petra van derKraak[5], Frederik Flenner[1,2], Bärbel M Ulmer[1,2], Ingke Braren[2,6], Xiaoke Yin[7], Konstantinos Theofilatos[7], Jorge Ruiz-Orera[8], Giannino Patone[8], Birgit Klampe[1,2], Thomas Schulze[1,2], Angelika Piasecki[1,2], Yigal Pinto[9], Aryan Vink[5], Norbert Hübner[8,10,11,12], Sian Harding[13], Manuel Mayr[7], Chris Denning[3], Thomas Eschenhagen[1,2] & Arne Hansen[1,2,*] (ID)

## Abstract

The phospholamban (PLN) p.Arg14del mutation causes dilated cardiomyopathy, with the molecular disease mechanisms incompletely understood. Patient dermal fibroblasts were reprogrammed to hiPSC, isogenic controls were established by CRISPR/Cas9, and cardiomyocytes were differentiated. Mutant cardiomyocytes revealed significantly prolonged $Ca^{2+}$ transient decay time, $Ca^{2+}$-load dependent irregular beating pattern, and lower force. Proteomic analysis revealed less endoplasmic reticulum (ER) and ribosomal and mitochondrial proteins. Electron microscopy showed dilation of the ER and large lipid droplets in close association with mitochondria. Follow-up experiments confirmed impairment of the ER/mitochondria compartment. PLN p.Arg14del end-stage heart failure samples revealed perinuclear aggregates positive for ER marker proteins and oxidative stress in comparison with ischemic heart failure and non-failing donor heart samples. Transduction of PLN p.Arg14del EHTs with the $Ca^{2+}$-binding proteins GCaMP6f or parvalbumin improved the disease phenotype. This study identified impairment of the ER/mitochondria compartment without SR dysfunction as a novel disease mechanism underlying PLN p.Arg14del cardiomyopathy. The pathology was improved by $Ca^{2+}$-scavenging, suggesting impaired local $Ca^{2+}$ cycling as an important disease culprit.

**Keywords** endoplasmic reticulum; engineered heart tissue; human-induced pluripotent stem cells; mitochondria; phospholamban p.Arg14del
**Subject Categories** Cardiovascular System; Genetics, Gene Therapy & Genetic Disease

## Introduction

Dilated cardiomyopathy (DCM) is a common cause of severe heart failure (HF). Mutations in several genes central to cardiomyocyte biology have been described to cause DCM (McNally & Mestroni, 2017). The disease relevance of mutations in the gene encoding for phospholamban (PLN) was first discovered in 2003 (Haghighi *et al*, 2003; Schmitt *et al*, 2003). PLN is a 6.1 kDa protein localized in the membrane of the sarcoplasmic and endoplasmic reticulum (SR/ER). It is regulated by protein kinase-mediated phosphorylation and acts as a brake on the sarcoplasmic/endoplasmic reticulum $Ca^{2+}$ ATPase (SERCA2). In its non-phosphorylated state, PLN inhibits $Ca^{2+}$ re-uptake into the SR/ER by forming a hetero-dimer with SERCA2. Upon phosphorylation by cAMP (PKA)- or cGMP (PKG)-dependent

1  Department of Experimental Pharmacology and Toxicology, University Medical Center Hamburg-Eppendorf, Hamburg, Germany
2  German Center for Heart Research (DZHK), Kiel, Germany
3  Division of Cancer & Stem Cells, Biodiscovery Institute, University of Nottingham, Nottingham, UK
4  Electron Microscopy Unit, Center for Molecular Neurobiology Hamburg, University Medical Center Hamburg-Eppendorf, Hamburg, Germany
5  Department of Pathology, University Medical Center Utrecht, Utrecht University, Utrecht, The Netherlands
6  Vector Core Unit, University Medical Center Hamburg-Eppendorf, Hamburg, Germany
7  King's British Heart Foundation Centre of Research Excellence, King's College London, London, UK
8  Cardiovascular and Metabolic Sciences, Max Delbrück Center for Molecular Medicine in the Helmholtz Association (MDC), Berlin, Germany
9  Department of Experimental Cardiology, Academic Medical Center, Amsterdam, The Netherlands
10  DZHK (German Centre for Cardiovascular Research), Berlin, Germany
11  Charité -Universitätsmedizin, Berlin, Germany
12  Berlin Institute of Health (BIH), Berlin, Germany
13  British Heart Foundation Centre of Research Excellence, NHLI, Imperial College London, London, UK
    *Corresponding author. Tel: +49 40 741057207; E-mail: ar.hansen@uke.de
    †These authors contributed equally to this work

protein kinase at serine 16 or $Ca^{2+}$/calmodulin-dependent protein kinase II (CaMKII) at threonine 17, PLN dissociates from SERCA2 and forms a homo-pentamer, resulting in increased SERCA2 $Ca^{2+}$ affinity and facilitated $Ca^{2+}$ re-uptake into the SR/ER. Hence, PLN acts as a key regulator at the interphase between $Ca^{2+}$ homoeostasis and force production. Overexpression of PLN in mice resulted in reduced SR $Ca^{2+}$ uptake, diminished $Ca^{2+}$ transient amplitude, and impaired left ventricular function (Kadambi et al, 1996). In contrast, heterozygous or homozygous PLN knockout mouse models demonstrated improved SR $Ca^{2+}$ import and increased contractility (Luo et al, 1994, 1996; Hoit et al, 1995), suggesting a close correlation between PLN protein abundance and contractile function in mice (Lorenz & Kranias, 1997).

In recent years, several human DCM mutations in *PLN* have been discovered. PLN p. Leu39 results in expression of a truncated unstable PLN protein. In cellular expression systems, the PLN p. Leu39stop mutation leads to a maximally active SERCA2a pump. Heterozygous and homozygous mutant carriers exhibit severe cardiomyopathy (Haghighi et al, 2003). Importantly, substantial species differences exist regarding the disease outcome of PLN deficiency. While beneficial effects resulted from PLN ablation in mice, the absence of PLN protein in patients was associated with severe cardiomyopathy (Haghighi et al, 2003). PLN p. Arg9Leu and PLN p. Arg9His mutations potentially led to altered interaction with PKA and abnormal phosphorylation of PKA substrate proteins (Medeiros et al, 2011). Finally, PLN p. Arg9Cys, PLN p. Arg25Cys, and PLN p. Arg14del were described to exert super-inhibitory effects on SERCA2a activity (Schmitt et al, 2003; Haghighi et al, 2012; Liu et al, 2015).

PLN p. Arg14del was first identified in a greek family (Haghighi et al, 2006) and subsequently identified as a DCM founder mutation in the Netherlands, where this mutation is accounting for up to 15% of all DCM cases (Van Der Zwaag et al, 2012). Clinically, PLN p. Arg14del mutation carriers present with left ventricular dysfunction and dilation and malignant ventricular arrhythmia. Transgenic overexpression of PLN p. Arg14del in wild-type (WT) mice resulted in the development of cardiomyopathy and reduced survival. SR preparations from PLN p. Arg14del overexpressing mice showed a small, but significant rightward shift of the SR $Ca^{2+}$-uptake rate, suggesting that PLN p. Arg14del, similar to the other mutant forms, acts as a super-inhibitor of SERCA2a (Haghighi et al, 2006). The effect was shown to depend on the presence of WT PLN, because transgenic overexpression of PLN p. Arg14del in a PLN knockout mouse model did not alter $Ca^{2+}$ re-uptake kinetics (Haghighi et al, 2012). Of note, normal SR $Ca^{2+}$ uptake in this model was associated with a similarly severe DCM phenotype, indicating that mechanisms distinct from abnormal SR $Ca^{2+}$ cycling contribute to the pathology of PLN p. Arg14del cardiomyopathy. Homozygous PLN p. Arg14del knock-in mice exhibited a DCM phenotype, with cardiac dilation and fibrosis, accompanied by contractile dysfunction, and arrhythmia (Eijgenraam et al, 2020).

It is difficult to elucidate disease mechanisms exerted by PLN mutations in mice. This is exemplified by the antithetic effects observed in response to complete PLN deletion, which relates to the lower chronotropic reserve of the mouse versus the human heart (Kranias & Hajjar, 2012). This limitation emphasizes the necessity for human-based studies to unravel disease mechanisms associated with PLN p. Arg14del. Histological analysis of explanted human hearts and left ventricular biopsies from PLN p. Arg14del heart

failure (HF) patients showed dense perinuclear globular PLN-positive aggregates as a disease-specific characteristic (te Rijdt et al, 2016, 2017). Human-induced pluripotent stem cell-derived cardiomyocytes (hiPSC-CMs) from a DCM patient carrying a PLN p. Arg14del mutation revealed a significantly higher spontaneous beating frequency, irregular $Ca^{2+}$ transients, and lower force development than respective isogenic controls (Karakikes et al, 2015; Stillitano et al, 2016). The authors concluded that the phenotype was compatible with super-inhibition of SERCA2a by PLN p. Arg14del. However, the higher caffeine-induced $Ca^{2+}$ transient amplitude in PLN p. Arg14del cardiomyocytes is difficult to reconcile with super-inhibition of SERCA2a and indicative of other, yet unknown consequences of PLN p. Arg14del.

The present study aimed at shedding new light on the molecular disease mechanisms evoked by the PLN p. Arg14del mutation and involved: (i) establishment of a patient-specific PLN p. Arg14del hiPSC line and a respective repaired isogenic control; (ii) phenotypical and functional characterization of hiPSC-CM in a three-dimensional engineered heart tissue (EHT) format showing preserved SR function; (iii) proteomic and RNA-seq analysis suggesting impairment of the ER/mitochondria compartment; (iv) confirmation by histology of end-stage HF patient and non-failing human heart samples; and (v) improvement of the disease phenotype by heterologous expression of $Ca^{2+}$ scavengers.

Taken together, the data suggest that PLN p. Arg14del cardiomyopathy is associated with functional alterations in the ER/mitochondria compartment with improvement by $Ca^{2+}$ scavenging.

## Results

### Clinical profile and hiPSC derivation

A female 31-year-old DCM patient was identified as a carrier of a heterozygous PLN p. Arg14del mutation. The patient was diagnosed with DCM at the age of 28 years and presented clinically with HF symptoms and ventricular arrhythmia. Her mother was also identified as a PLN p. Arg14del mutation carrier and diagnosed with DCM (Fig 1A). Patient-derived dermal fibroblasts were reprogrammed to hiPSC, and a respective isogenic control hiPSC line (PLNic) was generated by CRISPR/Cas9 technology (Fig 1B). Pluripotency was demonstrated by immunofluorescence (IF) staining for Oct-4A or TRA-1-60. G-banding demonstrated a normal karyotype (Appendix Fig S1A–E). No off-target effects after CRISPR/Cas9 genome editing were detected by sequencing the 10 most likely off-target loci (Appendix Fig S1F). Cardiomyocytes from PLN p. Arg14del (78 ± 15% positive for cTNT; mean ± SD, $n = 9$ batches from two clones) and PLNic hiPSC (80 ± 16% positive for cTNT; mean ± SD, $n = 9$ batches from two clones) were differentiated, and hiPSC-CMs were cultured in two-dimensional (2D) and three-dimensional (3D, engineered heart tissue; EHT) format. Quantitative RT–PCR revealed no detectable PLN p. Arg14del mRNA in PLNic hiPSC-CM and approximately 50% expression of both, PLN WT and p. Arg14del mRNA, in PLN p. Arg14del hiPSC-CM (Fig 1C). Analysis of total RNA from left ventricular samples obtained from two independent DCM patients carrying a heterozygous PLN p. Arg14del mutation revealed a ~1:1 ratio of PLN WT and p. Arg14del mRNA (Fig 1D). IF analysis of 2D hiPSC-CM revealed similar perinuclear

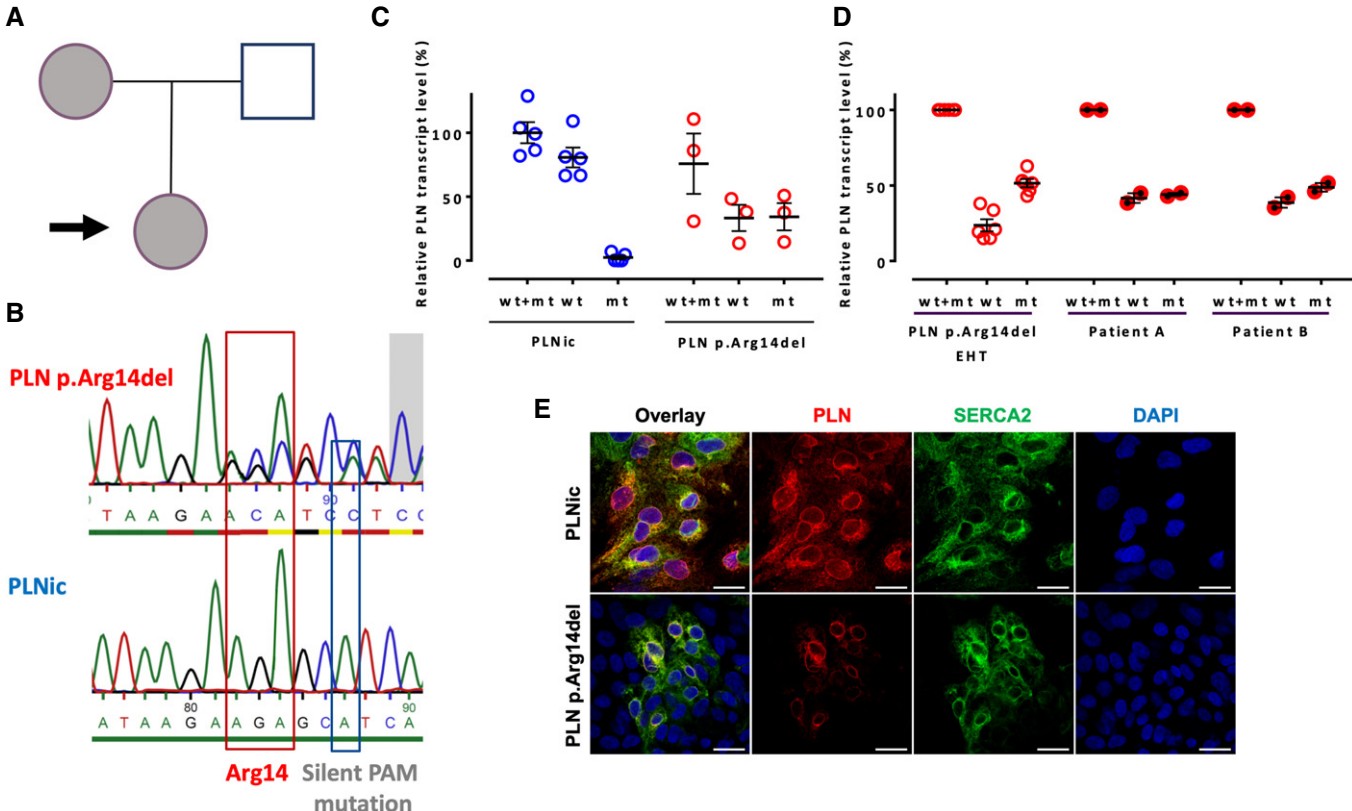

**Figure 1. HiPSC line derivation and validation.**

A    Pedigree of the patient family. Arrow: Index patient; index patient and mother are PLN p.Arg14del mutation carriers and were diagnosed with DCM.
B    Sequencing of PLN patient-derived hiPSC (PLN p.Arg14del) and isogenic controls (PLNic) to confirm CRISPR/Cas9-mediated gene correction. Red box indicates codon for Arg14, and blue box indicates silent PAM (protospacer adjacent motif) mutation.
C, D  Relative *PLN* transcript level (normalized to *ACTN2* and PLNic wt+mt) with PCR primers amplifying PLN wild type (wt) and p.Arg14del (mt), only wt or only p.Arg14del (mt). (C) Analysis of PLNic ($n = 9$ EHTs from 5 batches, each replicate consists of a pool of 1–2 EHTs from separate batches) and PLN p.Arg14del ($n = 9$ EHTs from 3 batches, each replicate consists of a pool of 2–4 EHTs from separate batches) EHTs, mean ± SEM. (D) Analysis of PLN p.Arg14del EHTs ($n = 9$ EHTs from 3 batches, each replicate consists of a pool of 2–4 EHTs from separate batches) and two independent failing heart patient samples from heterozygous PLN p.Arg14del mutation carriers (each $n = 1$ RNA sample, 2 separate PCR analysis), mean ± SEM.
E    Immunofluorescence of 2D hiPSC-CM from each genotype with antibodies against PLN (red) and SERCA2 (green) and DAPI staining of nuclei (blue); scale bar 20 μm.

PLN and SERCA2 protein localization in both genotypes, albeit staining was weaker in PLN p. Arg14del (Fig 1E, Appendix Fig S1G and H). Western blot analysis of EHTs showed lower abundance of monomeric PLN protein in PLN p. Arg14del and PLNic versus human non-failing heart (NFH), with no difference in the pentameric PLN form (Fig 2A and B). The close proximity between PLN p. Arg14 and PLN p. Ser16, the phosphorylation target site, raised the question, whether the response to the β-adrenoceptor agonist isoprenaline (ISO) was altered in PLN p. Arg14del. Western immunoblot analysis showed that ISO significantly enhanced band intensity of the pentameric pSer16 PLN only for PLNic, suggesting lower ISO-mediated phosphorylation of PLN p. Arg14del (Fig 2C and D). Functionally, ISO exerted a positive inotropic (Force PLNic: +61%, PLN p. Arg14del: +64%) and lusitropic effect (relaxation time PLNic: −7%, PLN p. Arg14del: −11%), without apparent differences between genotypes (Fig 2E and F, Appendix Fig S2A). This showed that despite lower ISO-mediated phosphorylation in PLN p.

Arg14del, the canonical inotropic and lusitropic force responses to ISO remained unchanged.

**Contractile force**

Contractile analysis in EHTs (6 differentiation batches; two different clones) revealed that PLN p. Arg14del EHTs developed significantly lower force and a higher beating frequency than PLNic assessed at two different time points (Fig 3A–D, Movies EV1 and EV2). Force analysis under electrical stimulation (2 Hz) confirmed lower forces of PLN p. Arg14del EHTs and showed no apparent differences in time to peak (TTP$_{-80\%}$) and relaxation time (RT$_{80\%}$) (Fig 3E, G, H and I). Normalized average contraction peaks revealed a significant prolongation of the relative relaxation time for PLN p. Arg14del (RT$_{80\%}$/force amplitude, +170%, Fig 3F and J). Overall, the data demonstrated that PLN p. Arg14del EHTs beat faster, develop 50% less force, and exhibit a late relaxation deficit.

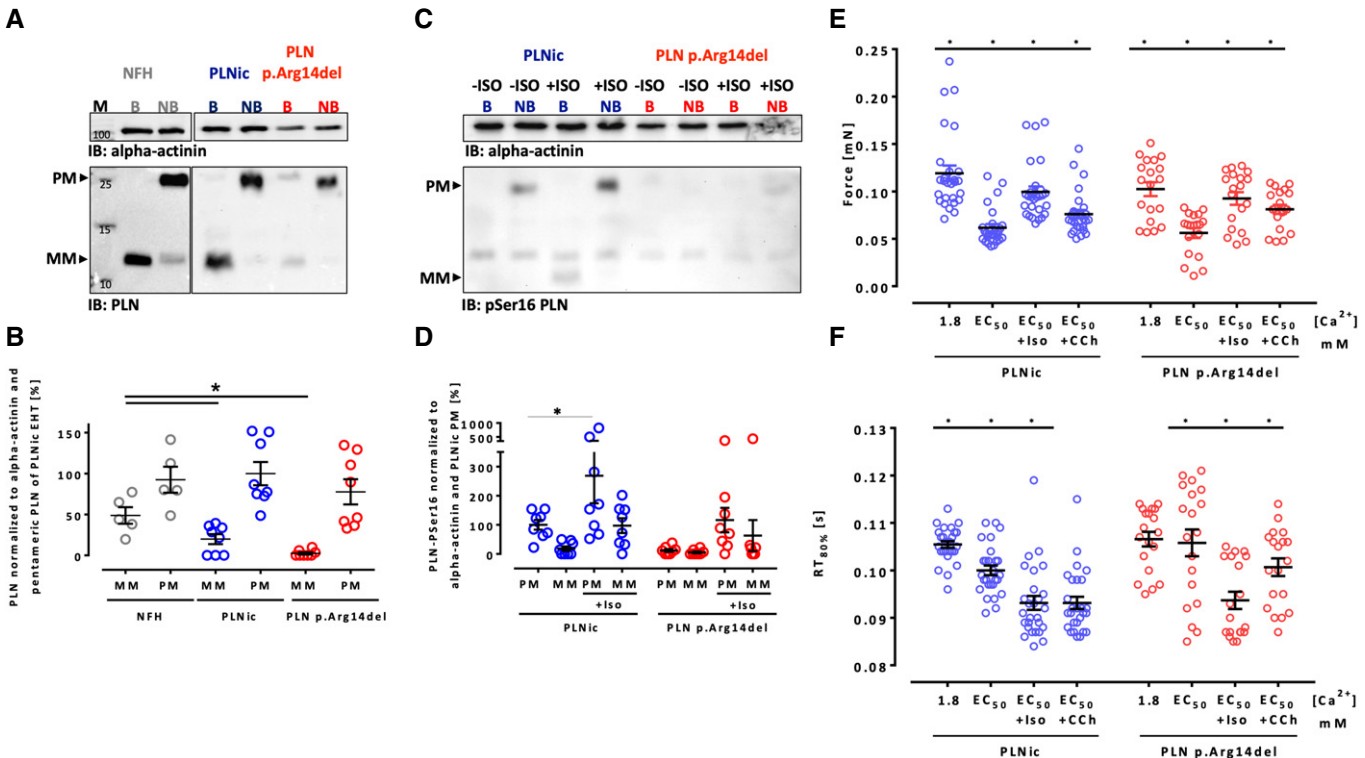

**Figure 2. PLN expression profile and response to isoprenaline.**

A    Western immunoblot analysis of total PLN in non-failing heart tissue (NFH), PLNic, and PLN p.Arg14del EHTs. Pentameric (PM) and monomeric (MM) form of PLN under boiled (B) and non-boiled (NB) conditions.

B    PLN protein quantification of PM and MM forms under non-boiled condition. NFH ($n = 5$ protein samples from two different donor hearts), PLNic ($n = 8$, each replicate consists of a pool of 3 EHTs from three different separate batches), and PLN p.Arg14del ($n = 8$, each replicate consists of a pool of 3–4 EHTs from three different separate batches). Loading was normalized to α-actinin. One-way ANOVA of PM and MM forms with Tukey's post-test, mean ± SEM, * $P < 0.05$.

C    Western immunoblot analysis of PLN pSer16 MM and PM in PLNic and p.Arg14del EHTs (non-boiled) in the absence and presence of isoprenaline (ISO; 100 nM, loading was normalized to α-actinin).

D    PLN pSer16 quantification of PM and MM form (non-boiled; loading was normalized to α-actinin). PLNic ± isoprenaline ($n = 8$, each replicate consists of a pool of 3–4 EHTs from three different separate batches) and PLN p.Arg14del ± isoprenaline ($n = 8$, each replicate consists of a pool of 3–4 EHTs from three different separate batches). One-way ANOVA of PM and MM forms with Sidak's post-test, * $P < 0.05$. Mean ± SEM.

E, F    Force (E) and relaxation time (RT) (F) of PLNic and PLN p.Arg14del EHTs in Tyrode's solution (Ca²⁺ 1.8 mM), Tyrode's solution (EC₅₀ [Ca²⁺]: PLNic: 0.4–0.6 mM, PLN p.Arg14del: 0.7–0.8 mM), in the presence of ISO (100 nM) or carbachol (10 μM); PLNic ($n = 27$ EHTs from 4 batches), PLN p.Arg14del EHTs ($n = 19$ EHTs from 3 batches), mean ± SEM, one-way ANOVA for PLNic or PLN p.Arg14del EHTs with Tukey's post hoc test, * $P < 0.05$. The data are representative of $n = 6$ independent experiments.

## Ca²⁺ transient analysis

Analysis of Ca²⁺ transients in Fura 2-loaded 2D hiPSC-CM showed a significant prolongation of late Ca²⁺ decay time (DT) in PLN p. Arg14del (DT₈₀% PLNic: $0.337 ± 0.01$ s, $n = 25$; PLN p. Arg14del: $0.437 ± 0.02$ s, $n = 24$). Notably, this was not associated with differences in Ca²⁺ transient amplitude, diastolic Ca²⁺ content, or Ca²⁺ transient decay tau (Fig 4A–E), nor was the caffeine-induced Ca²⁺ transient amplitude altered (Appendix Fig S2B–F). The late DT prolongation was compatible with the late relaxation deficit, but normal caffeine-induced Ca²⁺ transient amplitude argued against gross abnormalities of SR function in PLN p. Arg14del EHTs.

## Ca²⁺ induced irregular beating pattern

In patients, PLN p. Arg14del DCM is often accompanied by severe arrhythmia. Indeed, PLN p. Arg14del displayed an occasional irregular beating pattern (IBP) under standard cell culture conditions (Fig 4F). To allow quantification and to study the dependence of IBP on Ca²⁺ loading, EHT contractility was recorded for 9 h in Tyrode's solution supplemented with 1.0, 1.8, or 3.0 mM extracellular Ca²⁺. RR scatter (interdecile range of mean beat-to-beat distance) in PLNic EHTs was significantly higher at 1.8 mM than 1.0 mM Ca²⁺ and showed a Ca²⁺ concentration-dependent increase in IBP in PLN p. Arg14del EHTs. The RR scatter at 1.8 and 3.0 mM was significantly higher in PLN p. Arg14del than in PLNic EHTs (Fig 4G), corroborating a Ca²⁺-dependent IBP-phenotype in PLN p. Arg14del.

## Proteomic analysis, quantification, and function of mitochondria, ultrastructural analysis

Given the apparent discrepancy between severe contractile dysfunction and IBP in PLN p. Arg14del despite the absence of changes in intracellular Ca²⁺ transients and SR Ca²⁺ loading, EHTs

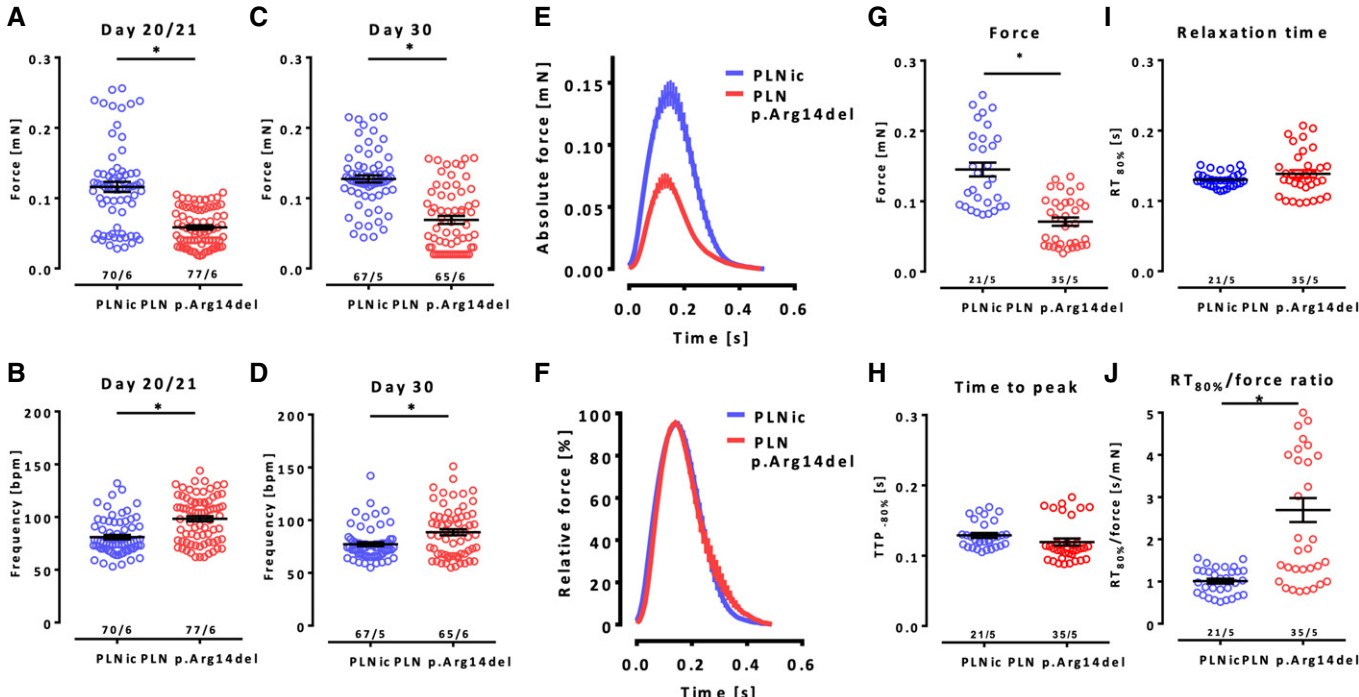

**Figure 3. Analysis of contractile force in EHTs.**

A–D Force and frequency analysis of spontaneous force development and beating frequency in EHTs on days 20/21 (A, B) and day 30 (C, D) of EHT development, PLNic: $n = 70/67$ EHTs from 5–6 batches (as indicated in the figure), PLN p.Arg14del: $n = 77/65$ EHTs from 6 batches, mean ± SEM, Mann–Whitney $U$-test, $*P < 0.05$. The data are representative of $n = 10$ independent experiments for days 20/21 and $n = 9$ independent experiments for day 30. Force: days 20/21 PLNic: 0.11 ± 0.007 mN; p.Arg14del: 0.06 ± 0.003 mN; and day 30 PLNic: 0.13 ± 0.005 mN; p.Arg14del: 0.07 ± 0.005 mN; frequency: days 20/21 PLNic: 81 ± 2 BPM; p.Arg14del: 98 ± 2 BPM; and day 30 PLNic: 77 ± 2 BPM; p.Arg14del: 88 ± 3 BPM.

E–J Contractility analysis of electrically stimulated EHTs (2 Hz) on days 21 ± 3 of EHT development. Absolute (E) and relative (F) (normalized to peak maximum) average contraction peaks. Analysis of force (G), time to peak (H), relaxation time (I), and relaxation time/force ratio (J), PLNic: 0.14 ± 0.009 mN, $n = 21$, 5 batches; PLN p.Arg14del: 0.07 ± 0.005 mN, $n = 35$, 5 batches, mean ± SEM, unpaired two-sided Student's $t$-test, $*P < 0.05$. The data are representative of $n = 8$ independent experiments.

of both genotypes were subjected to label-free proteomic analysis in the search for alternative mechanisms. In total, 2,843 proteins were detected and 1,576 proteins differed significantly between PLNic and PLN p. Arg14del (FDR < 0.05; Dataset EV1). Principal component analysis showed separate clustering of PLNic from PLN p. Arg14del for all replicates (Fig 5A). Unsupervised hierarchical clustering confirmed differences between PLNic and PLN p. Arg14del (Appendix Fig S3A). Enrichment pathway analysis identified differences in KEGG pathways related to ribosome, protein processing in endoplasmic reticulum (ER), lysosome, phagosome, peroxisome, glutathione metabolism, and energy metabolism/mitochondria between PLNic and PLN p. Arg14del (Fig 5B and C; Tables EV1 and EV2). Differently abundant ER proteins included $Ca^{2+}$-binding chaperones and proteins involved in protein folding, N-glycosylation, and ubiquitination. Congruent with the observed alteration of mitochondrial proteins, quantitative analysis of mitochondrial DNA (mtDNA, Mt-ND1, 2) demonstrated 3.3-fold lower values for PLN p. Arg14del (Appendix Fig S3B). Co-expression network analysis of metabolism-related proteins revealed over-representation of carbohydrate, fatty acid, lipid, and one-carbon substrate metabolism in PLN p. Arg14del (Appendix Fig S3C), supported by differential abundance of metabolic key enzymes

such as 5′ AMP-activated protein kinase (AMPK) and glycogen phosphorylase (Dataset EV1).

Interestingly, constituent proteins of the ER/mitochondria contact site that ascertain unperturbed ER-mitochondrial crosstalk, namely calnexin (Lynes et al, 2013; Bravo-Sagua et al, 2016), the voltage-dependent anion-selective channel (Szabadkai et al, 2006), and mitofusin-2 (Papanicolaou et al, 2011; Chen et al, 2012), were among the lower abundant proteins in PLN p. Arg14del. Also PLN, but not SERCA, was lower abundant in PLN p. Arg14del, supporting our previous IF data (Fig 1E; Appendix Fig S3D–K). To investigate whether differences in mitochondrial number and shape were detectable in PLN p. Arg14del, ultrastructural analysis of hiPSC-CM EHTs was performed. Indeed, transmission electron microscopy (TEM) confirmed lower abundance of mitochondria in PLN p. Arg14del, large lipid droplets in close association with mitochondria and pathologically dilated ER structures (Fig 5D and E). Accumulation of perinuclear lipid droplets in PLN p. Arg14del was subsequently confirmed by IF analysis of 2D hiPSC-CM by BODIPY™ staining (Fig 5F) and suggests disturbed lipid transfer via ER/mitochondria contact sites (Janer et al, 2016).

Western immunoblot analysis was used as alternative methodology to validate the differential abundance of candidate proteins

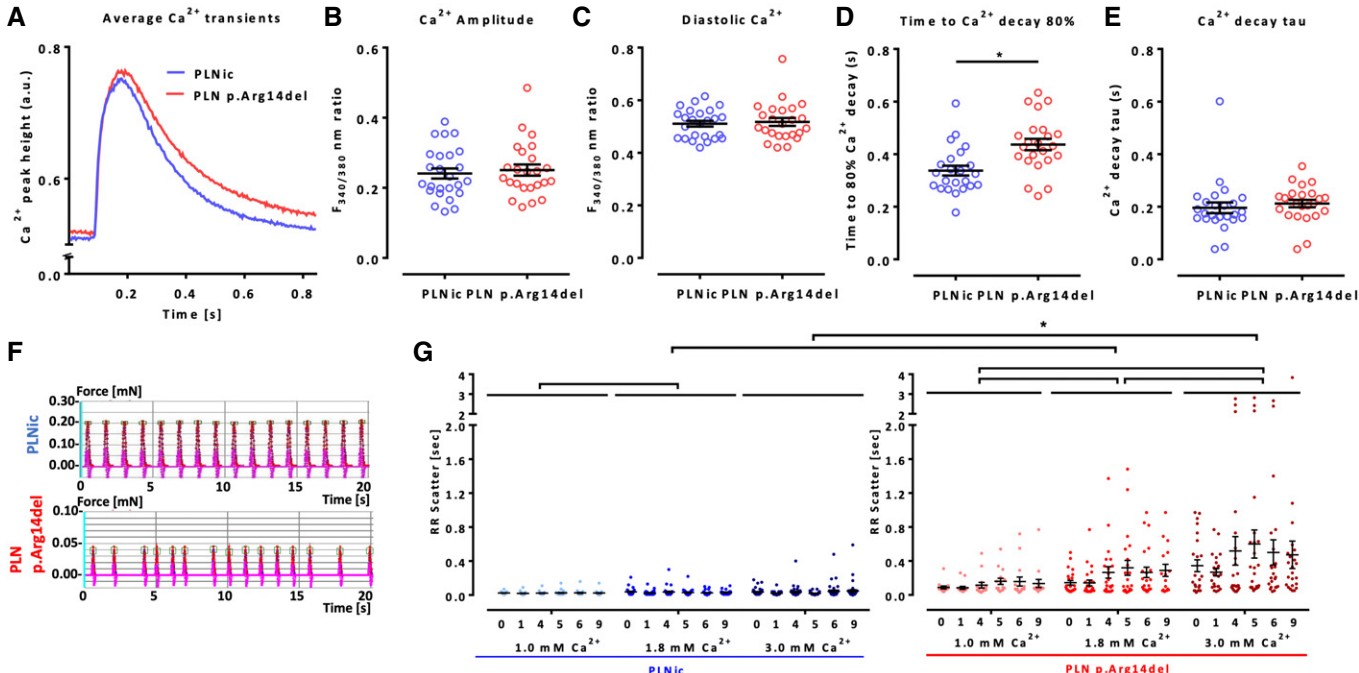

**Figure 4. Ca²⁺ transient analysis in Fura 2-loaded 2D hiPSC-CM.**

A–E Average Ca²⁺ transient peak (A), Ca²⁺ transient amplitude (B), diastolic Ca²⁺ (C), time to decay 80% (D), and Ca²⁺ decay tau (E); PLNic: $n = 25$ hiPSC-CM from 2 batches and PLN p.Arg14del: $n = 24$ hiPSC-CM from 2 batches, mean $\pm$ SEM, Mann–Whitney U-test, *$P < 0.05$.

F Representative EHT recording.

G RR scatter (surrogate for irregular beating pattern, IBP, interdecile range of mean beat-to-beat distance) of PLNic and PLN p.Arg14del EHT at 1.0, 1.8, and 3.0 mM extracellular Ca²⁺ concentration, incubation time 9 h, recording time 50 s, PLNic: 1.0 mM: $n = 24/3$, 1.8 mM: $n = 32/4$, 3.0 mM: $n = 41/5$ (EHT/batches), PLN p.Arg14del: 1.0 mM: $n = 15/2$, 1.8 mM: $n = 24/3$, 3.0 mM: $n = 23/3$ (EHT/batches), mean $\pm$ SEM, two-way ANOVA (comparing hiPSC lines and Ca²⁺ concentrations (but not time points)) with Tukey's post hoc test, *$P < 0.05$. The data are representative of $n = 5$ independent experiments.

observed in the proteomic screen. Signal intensities for galectin-3 and SAFB-like transcription modulator (SLTM) were significantly higher in PLN p. Arg14del. LIM and cysteine-rich domains 1 (LMCD1) and calnexin exhibited weaker band intensities in PLN p. Arg14del. No difference was detected for reticulocalbin-3. In addition, phosphorylation of pyruvate dehydrogenase (PDH) at pS293 indicative of enzyme inhibition and phosphorylation of AMPK at pThr172 indicative of enzyme activation were higher in PLN p. Arg14del, pointing to alteration in ER-mitochondria crosstalk (Fig EV1A and B). The reduction in calnexin protein levels shown by Western immunoblot analysis was supported by weaker IF staining and a loss of the perinuclear localization of calnexin in PLN p. Arg14del 2D hiPSC-CM (Fig EV1C).

In order to investigate whether the reduced mitochondrial content and the discrepant metabolic profile in PLN p. Arg14del were paralleled by impairment of mitochondrial respiration, Seahorse experiments were performed in 2D hiPSC-CM. PLN p. Arg14del revealed lower oxygen consumption rate under baseline and experimental conditions, resulting in lower values for basal and maximal respiration rate and lower ATP production (Fig 6A–D). In line with this, fluorometric detection of the oxidation of a dichloro-hydrofluorescein probe revealed significantly higher steady state levels of reactive oxygen and nitrogen species (ROS/RNS) in PLN p. Arg14del (Fig 6E). IF staining of 2D hiPSC-CM of both genotypes with an established marker of oxidative stress, 8-hydroxydeoxyguanosine

(8-OHdG), revealed enhanced signal intensity in PLN p. Arg14del (Fig 6F). This observation was supported by enhanced detection of derivatized cellular protein carbonylation in PLN p. Arg14del by Western immunoblotting (Fig 6G). This finding supports our proteomic data that suggested a stronger representation of the KEGG pathway for glutathione metabolism (Table EV1) and a higher abundance of superoxide dismutase (SOD1) in PLN p. Arg14del (Dataset EV1) suggesting oxidative stress as a confounder in PLN p. Arg14del cardiomyopathy.

### Transcriptomic analysis

To investigate differences in the transcriptomic profile of PLNic versus PLN p. Arg14del during the differentiation to hiPSC-CM, RNA sequencing analysis was performed at specific stages of the differentiation procedure. Principal component analysis on day 0 (undifferentiated hiPSC), day 3 (mesodermal progenitors), day 8 (cardiac progenitors), and day 15 (early CM) revealed close clustering of both genotypes according to the time point of differentiation (Appendix Fig S4C). Analysis of prototypical differentiation markers (Fu et al, 2018) (Branco et al, 2019) revealed stage-specific marker expression in both genotypes, but lower expression of cardiac markers in PLN p. Arg14del on day 15 (Appendix Fig S4A and B). Accordingly, a number of cardiac KEGG pathways were enriched in PLNic on day 15 (Appendix Fig S4D and E). Notably, ER/

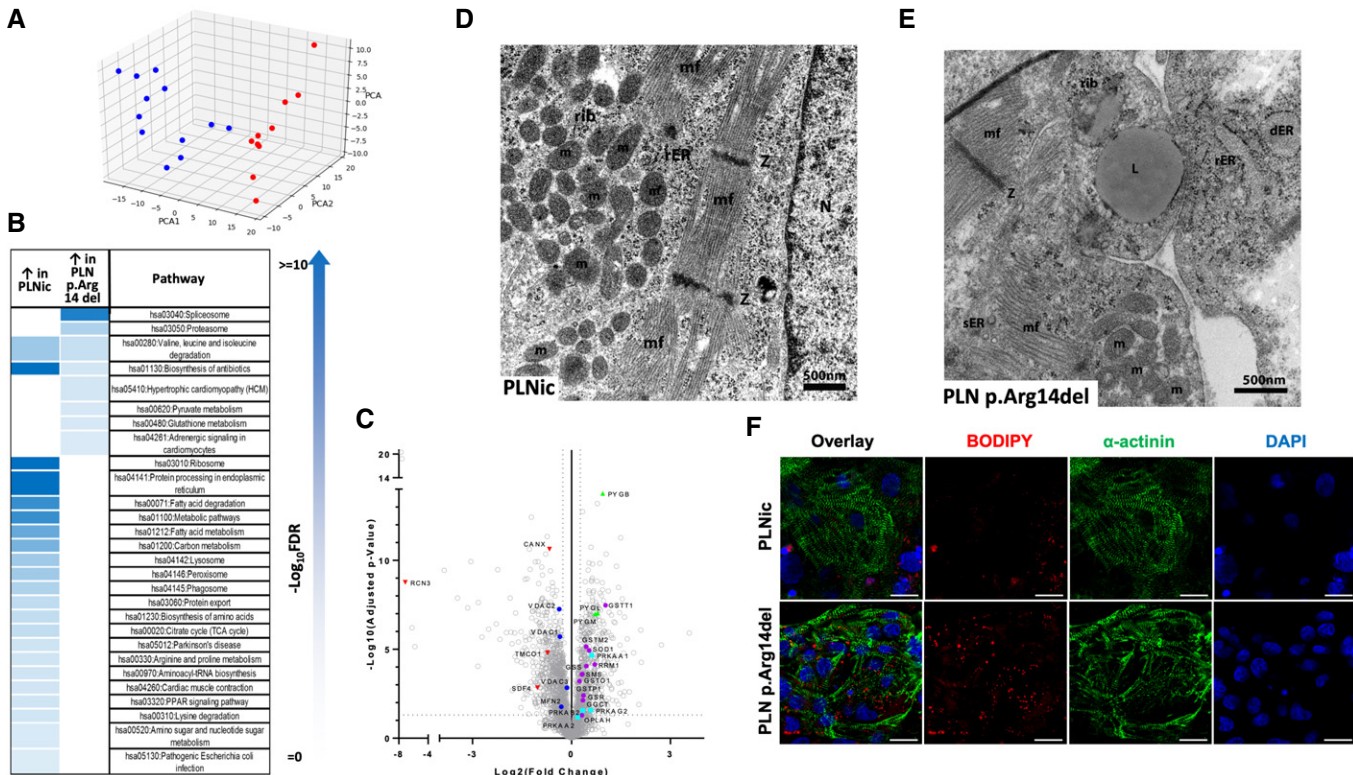

**Figure 5. Proteomic analysis, transmission electron microscopy.**

A    Principal component analysis (PCA) of PLNic (blue, *n* = 12) and PLN p.Arg14del (red, *n* = 10) EHTs based on their proteomic profiles. Each dot represents one EHT.
B    Significantly enriched pathways of up- and downregulated proteins.
C    Volcano plot of log2 fold changes (PLNic versus PLN p.Arg14del) and log10 of the *P*-values with color-coded significance levels.
D, E  Transmission electron microscopy of PLNic and PLN p.Arg14del EHTs; mf: myofilaments, Z: Z-line, rib: ribosome, m: mitochondria, sER: smooth endoplasmic reticulum, rER: rough endoplasmic reticulum, dER: dilated endoplasmic reticulum, N: nucleus, and L: lipid droplet.
F    Immunofluorescence analysis of 2D hiPSC-CM from PLNic and PLN p.Arg14del with BODIPY™ lipid staining (red), α-actinin (green), and DAPI staining for nuclei (blue); scale bar 20 μm.

mitochondria contact site markers (e.g., *CANX*, *VDAC 1-3*, *RCN-3*, *MFN2*) showed lower expression and *ITPR2* showed higher expression in PLN p. Arg14del on day 15 (Appendix Fig S5A–I). ER stress response and fetal gene program marker genes did not differ between genotypes.

**Immunohistochemistry in human heart failure samples**

Immunohistochemistry was employed to investigate whether defects in the ER/mitochondria compartment that were identified in the present study were also relevant and typical for human PLN p. Arg14del end-stage HF patients. Autopsy and explanted left ventricular samples from NFH, ischemic HF (IHF), and PLN p. Arg14del-related end-stage HF hearts were analyzed (Fig 7). PLN staining revealed the presence of perinuclear aggregates for PLN p. Arg14del, as previously reported (te Rijdt *et al*, 2016, 2017). RCN-3, an ER $Ca^{2+}$-binding chaperon protein that acts as a negative regulator of collagen production (Tsuji *et al*, 2006; Martínez-Martínez *et al*, 2017), was low abundant in PLN p. Arg14del (Dataset EV1). PLN end-stage HF showed significantly more RCN-3-positive perinuclear aggregates than NFH or IHF samples (Fig 7A and C), suggesting that

perinuclear aggregates linked to the ER. Notably, also for 8-OHdG, significantly more dotted perinuclear staining was detected in PLN p. Arg14del (Fig 7A and B). Two other dysregulated proteins in PLN p. Arg14del EHTs, which were not localized to the ER/mitochondrial compartment, did not show this perinuclear staining pattern: LMCD1, a sarcomeric Z-disk hypertrophy-mediating protein (Frank *et al*, 2010), was downregulated in PLN p. Arg14del EHTs and revealed an enhanced mosaic-like pattern with different cellular staining intensities particularly in PLN and to a lesser extent in IHF. SLTM, a pro-apoptotic nuclear protein (Chan *et al*, 2007), that was more abundant in PLN p. Arg14del EHTs revealed a nuclear staining pattern in PLN p. Arg14del, non-failing and IHF samples (Fig 7A). The perinuclear staining pattern of RCN-3 and 8-OHdG suggests ER alterations associated with oxidative stress in PLN p. Arg14del end-stage HF hearts.

**Ca²⁺ scavenging**

Complementary to $Ca^{2+}$ transient measurements in 2D hiPSC-CM (Fig 4A–D), analysis was complemented in 3D EHT format after transduction with the genetically encoded $Ca^{2+}$ sensor GCaMP6f.

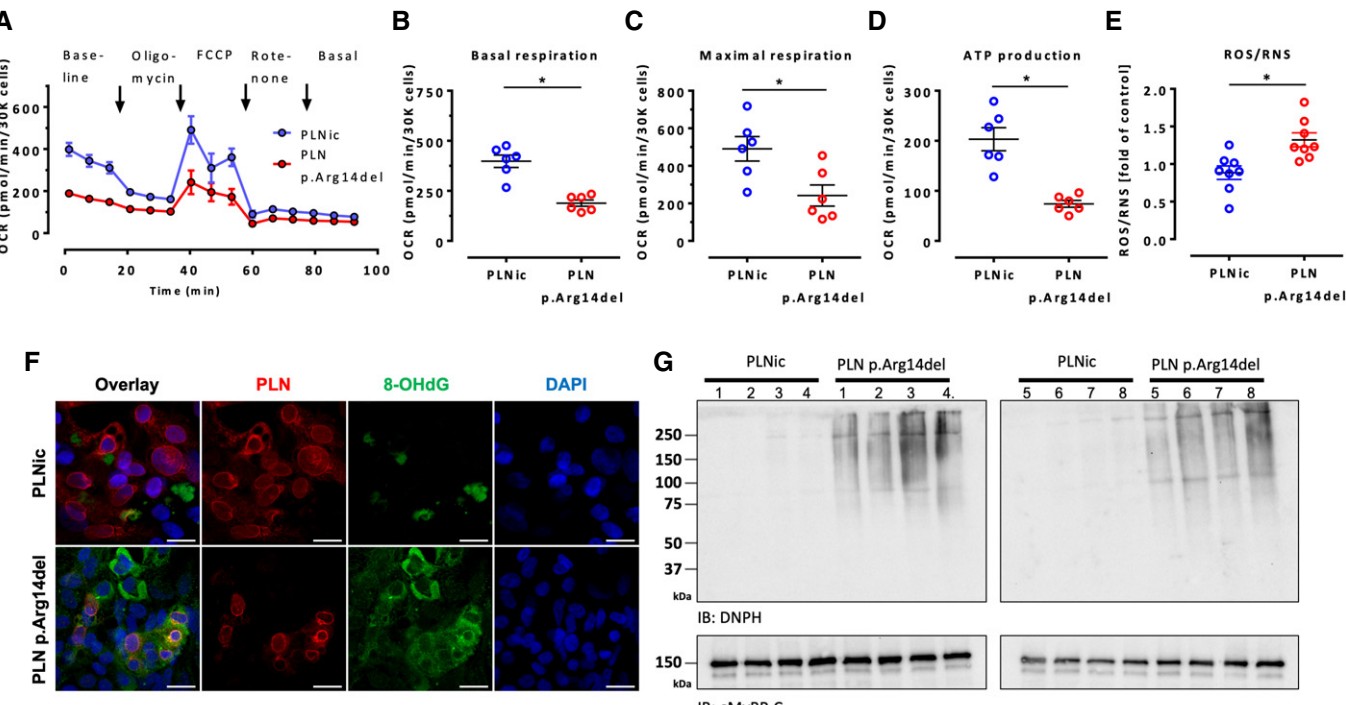

**Figure 6. Mitochondrial respiration and detection of oxidative stress.**

A   Oxygen consumption rate in PLNic versus PLN p.Arg14del. Mean ± SEM.

B–D   Quantification of basal respiration (B), maximal respiration (C) and ATP production (D). Mean ± SEM, $n = 6$ biological replicates (each biological replicate represents the average of 12 wells of a 96-well Seahorse plate), Mann–Whitney U-test, *$P < 0.05$.

E   Fluorometric detection of ROS content in complete medium supernatant or lysed cells after 24 h of EHT culture, $n = 8$ EHTs, each circle: Mean of 6 technical replicates, mean ± SEM, Mann–Whitney U-test, *$P < 0.05$.

F   Immunofluorescence of 2D hiPSC-CM from PLNic and PLN p.Arg14del for PLN (red), 8-hydroxydeoxyguanosine (8-OHdG; green) and DAPI-stained nuclei; scale bar 20 µm.

G   Western immunoblot analysis of 2D hiPSC-CM from PLNic and PLN p.Arg14del with an anti-2,4-dinitrophenylhydrazine antibody (DNPH) that detects the presence of carbonyl groups modified by 2,4-dinitrophenylhydrazine as a surrogate marker of irreversibly oxidized proteins, $n = 8$ replicates per genotype, cMyBP-C was used as a loading control.

Similarly, no difference in Ca$^{2+}$ transient amplitude was detected (Appendix Fig S6A–J). For PLNic, force development was slightly lower in the presence of GCaMP6f, an expected consequence due to its function as a Ca$^{2+}$ buffer. Surprisingly, contractile force improved for PLN p. Arg14del in the presence of GCaMP6f (PLN p. Arg14del force without GCaMP6f: 49% of PLNic value, Fig 3E; PLN p. Arg14del force with GCaMP6f: 81% of PLNic value, Appendix Fig S6F). This chance-finding suggested functional improvement of the PLN p. Arg14del phenotype by GCaMP6f-mediated Ca$^{2+}$ scavenging.

To support this observation, experiments were repeated using another Ca$^{2+}$-binding protein, the EF-hand protein parvalbumin (Wang & Metzger, 2008). HiPSC-CMs in 2D and 3D format were transduced with an empty control-, GCaMP6f-, or parvalbumin-encoding virus. IF analysis of 2D hiPSC-CM suggested no difference in PLN and SERCA2 protein expression nor localization and confirmed heterologous expression of both Ca$^{2+}$ scavenging proteins. While heterologous expression of GCaMP6f localized uniformly throughout the cell, a large fraction of the expressed parvalbumin resided in the nucleus (Appendix Fig S7A–E). In the control-transduced groups, phosphorylation of PDH at pS293 and AMPK at pThr172 was significantly higher in PLN p. Arg14del than in PLNic

(Fig 8A–D). In PLN p. Arg14del, expression of GCaMP6f or parvalbumin significantly reduced phosphorylation levels compared to control-transduced hiPSC-CM EHTs. Band intensities for PLN and SERCA2 remained unaltered (Appendix Fig S8A–D). Expression of GCaPM6f in PLN p. Arg14del was associated with significantly higher force development, but not for parvalbumin (Fig 8E). The small effect on force development in response to parvalbumin expression could be explained by the mainly nuclear localization of parvalbumin. Analysis of ER/unfolded protein response (UPR) stress response markers revealed higher gene expression of protein disulfide isomerase family A member 4 (PDIA4) and the ER chaperones BIP and HYOU1 in PLN p. Arg14del. BIP and HYOU1 expression was reduced by Ca$^{2+}$ scavenging in PLN p. Arg14del (Fig 8F–H). Other prototypical ER/UPR stress markers or fetal gene program marker did not reveal convincing differences between genotypes (Appendix Fig S8E–G, Appendix Fig S9A–D). This observation was in conjunction with the respective proteomic data (Appendix Fig S9E–G). IF staining for 8-OHdG revealed no signals for PLNic in all virus groups. In contrast, enhanced 8-OHdG signals were detectable for PLN p. Arg14del with no impact in response to Ca$^{2+}$ scavenging (Fig 8I and J). Proximity ligation assays (PLAs) were performed to

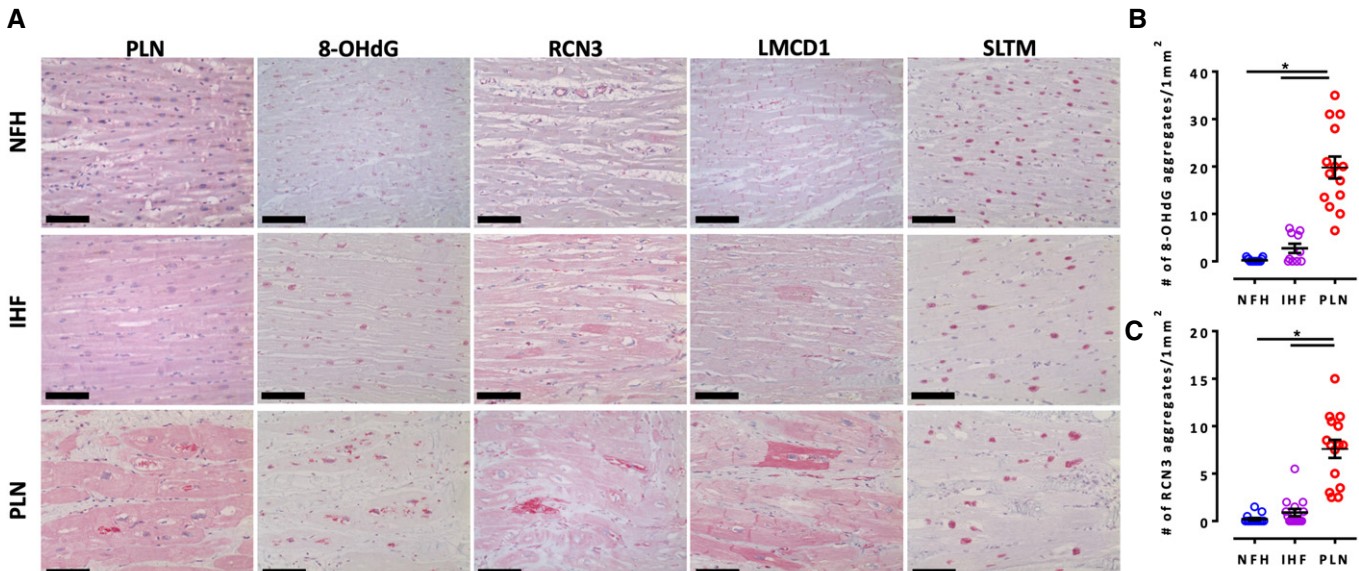

**Figure 7. Immunohistochemistry of non-failing heart (NFH), ischemic cardiomyopathy heart failure (ICM), and PLN p. Arg14del end-stage heart failure (PLN).**

A    8-Hydroxy-2′-deoxyguanosine (8-OHdG), reticulocalbin-3 (RCN3), LIM and cysteine-rich domains protein 1 (LMCD1), and SAFB-like transcription modulator (SLTM). Scale bar 100 μm.

B, C    Quantification of aggregates for 8-OHdG staining (B) (n = 3 different human hearts for NFH and PLN, n = 2 human hearts for IHF, 4–5 areas per heart) and RCN-3 (C) (n = 3 different human hearts for NFH, IHF, and PLN, 4–5 areas per heart) sections. One-way ANOVA with Tukey's post hoc test for multiple comparison, *P < 0.05, mean ± SEM.

investigate the proximity of ER-mitochondrial contact sites by using a combination of antibodies that detect VDAC1 that localizes to the outer mitochondrial membrane and IP₃R that localizes to the ER. This revealed no differences in PLNic between control-, GCaMP6f-, or parvalbumin-transduced hiPSC-CM (Fig EV2A–C). Thapsigargin exposure that provokes a stress response led to an increase in the PLA signal intensity for each condition in the PLNic group. The PLN p. Arg14del control-transduced hiPSC-CM revealed similar signal intensity as the PLNic control group. Transduction with the GCaMP6f or parvalbumin virus increased PLA signal intensity substantially, suggesting dynamic alterations within the ER-mitochondrial compartment. Interestingly, stress induced by thapsigargin exposure decreased PLA signal intensity for all PLN p. Arg14del conditions, in line with a lower tolerance to ER stress compared to PLNic (Fig EV2A and B). TEM analysis for the different virus conditions confirmed low abundance of mitochondria and the appearance of dilated and elongated structures of the ER in PLN p. Arg14del. In PLNic, no overt morphological differences in cellular organelles were detectable. In contrast, PLN p. Arg14del showed higher number of mitochondria after heterologous expression of GCaMP6f or parvalbumin. Additionally, in GCaMP6f expressing PLN p. Arg14del EHTs, no dilated ER structures were detectable (Fig EV3).

In aggregate, PLN p. Arg14del cardiomyopathy was successfully modeled in hiPSC-CM with reproduction of disease-specific features. This allowed us to study and subsequently decipher a novel pathomechanism of PLN p. Arg14del cardiomyopathy. Key aspects were validated in human end-stage PLN p. Arg14del HF samples. Improvement by Ca²⁺ scavenging points to the relevance of Ca²⁺ handling abnormalities.

## Discussion

Heterozygous PLN p. Arg14del mutation is an established cause of DCM and severe HF. The molecular mechanisms that govern disease development are subject of debate and incompletely understood. Super-inhibitory effect of PLN p. Arg14del on SERCA2a function was suggested as the major underlying molecular disease mechanism, despite the observation that PLN p. Arg14del overexpression in PLN KO mice induced a DCM phenotype without affecting SR function (Haghighi et al, 2012). The present study showed that SR function of hiPSC-CM or derived EHTs was unaffected by the presence of PLN p. Arg14del as reflected by unchanged Ca²⁺ transient amplitude, caffeine-induced Ca²⁺ release, and a canonical lusitropic response to ISO. Small prolongation of relaxation and Ca²⁺ decay time in PLN p. Arg14del hiPSC-CM or EHTs were observed, but the magnitude of the changes did not match the large decrease in contractile force and the Ca²⁺-dependent beating irregularities as a surrogate of arrhythmia. These observations raised the question, which alternative mechanisms could connect small defects in late Ca²⁺ removal from the cytosol and Ca²⁺ induced IBP with low force development. Results from the present study proposed alteration of the ER and mitochondria, altered post-translational modifications of key metabolic enzymes such as PDH and AMPK, perinuclear lipid accumulation, oxidative stress, mitochondrial dysfunction, and degeneration. These molecular abnormalities resembled a pathological pattern previously reported to reflect disturbed ER/mitochondria contact sites. Constitutive IP₃R-mediated Ca²⁺ release from the ER to the mitochondria ascertains unperturbed mitochondrial function and cellular bioenergetics. The key enzyme in the mitochondria that catalyzes the formation of acetyl-CoA as well as flavin and

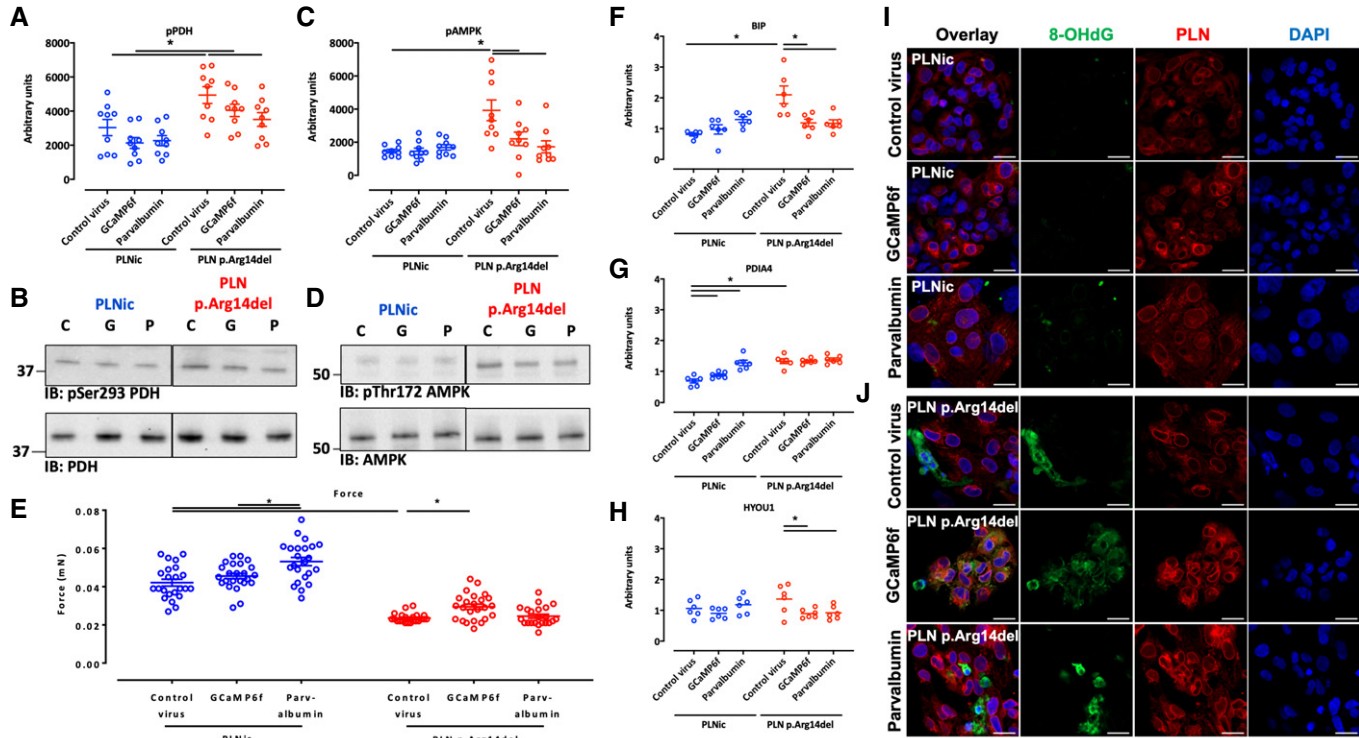

**Figure 8. Effect of Ca²⁺-scavenging.**

A–D Western immunoblots for pAMPK at Thr172 and pPDH at Ser293 for PLNic and PLN p.Arg14del transduced with control, GCaMP6f or parvalbumin virus, $n = 6–9$, each replicate consists of a pool of 2–3 EHTs. Two-way ANOVA (comparing genotypes and virus conditions) with Šidák's post-test, mean ± SEM, $*P < 0.05$. C: control virus, G: GCaMP6f virus, and P: parvalbumin virus.

E Force of PLNic and PLN p.Arg14del spontaneous beating EHTs on day 30 of EHT development, replicate numbers are PLNic: control virus: $n = 24$, GCaMP6f: $n = 23$, and parvalbumin: $n = 24$; and PLN p.Arg14del: control virus: $n = 22$, GCaMP6f: $n = 24$, and parvalbumin: $n = 22$; EHTs were from 2 different batches. Two-way ANOVA (comparing genotypes and virus conditions) with Šidák's post-test, mean ± SEM, $*P < 0.05$.

F–H Quantitative PCR of ER/UPR stress marker genes. PLNic and PLN p.Arg14del EHT after transduction with a control virus, or a virus encoding for GCaMP6f or parvalbumin; $n = 6$; each replicate consists of a pool of 2–3 EHTs. Two-way ANOVA (comparing genotypes and virus conditions) with Šidák's post-test, mean ± SEM, $*P < 0.05$.

I, J Immunofluorescence of 2D hiPSC-CM from PLNic and PLN p.Arg14del after transduction with control, GCaMP6f, or parvalbumin virus with antibodies recognizing PLN (red), 8-OHdG (green), and DAPI staining for nuclei (blue); scale bar 20 μm.

Source data are available online for this figure.

nicotinamide adenine dinucleotide from pyruvate is PDH. PDH activity is inhibited by phosphorylation via pyruvate dehydrogenase kinase 1, when IP₃-receptor-mediated Ca²⁺ current is reduced (Cárdenas *et al*, 2010). Indispensable for the integrity of the Ca²⁺-flux to sustain the crosstalk between ER and mitochondria is the close proximity between these organelles. Various experimental studies provided evidence that mitofusin-2 functions as a key component that tethers the mitochondria to the ER, thereby ensuring functional organelle linkage. Alteration of ER/mitochondria contact sites in mouse cardiomyocytes corroborated these mechanistic observations and described oxidative stress, mitochondrial insufficiency and degradation, hypo-contractility and cardiomyopathy with preserved Ca²⁺ transient amplitude as an immediate result (de Brito & Scorrano, 2008; Papanicolaou *et al*, 2011; Chen *et al*, 2012; Chen & Dorn, 2013; Song *et al*, 2014). Several features of ER/mitochondria contact site disruption were identified in the present study: reduced abundance of constituent components of the contact site, stronger phosphorylation of PDH, perinuclear lipid accumulation,

and oxidative stress. Putting this in context with the established function of SERCA2/PLN in both, SR and ER, this impairment of inter-organelle communication between ER and mitochondria argues strongly for a previously ignored contribution of the ER to the PLN p. Arg14del cardiomyopathy.

The ER is an extension of the nuclear membrane and shows a perinuclear subcellular localization. Therefore, it is not surprising to observe perinuclear PLN-positive aggregates in PLN p. Arg14del end-stage HF samples that co-localize with the ER (te Rijdt *et al*, 2016, 2017) and that also stained positive for RCN-3. Notably, TEM performed in HF samples provided evidence that the aggregates are in fact aggresomes and contain misfolded proteins when the protein degradation system of the cell is overwhelmed (te Rijdt *et al*, 2016, 2017). The presence of aggresomes thus reflects ER dysfunction (Scior *et al*, 2016), consequently supporting our findings.

The close interaction between ER and mitochondria is one important aspect of the current understanding of ER function via membrane contact sites (MCS) (Phillips & Voeltz, 2016). Other

important organelles that interact via ER-MCS are endosomes and Golgi vesicles, which are involved in the formation of lysosomes, phagosomes, and peroxisomes, also contributing to the degradation of misfolded proteins. Notably, reduced representation of KEGG pathways for lysosome, phagosome, and peroxisome pathways emphasized that the ER-MCS with these organelles could also be affected and thus further contribute to the deterioration of the PLN p. Arg14del cardiomyopathy. The relevance of these findings is strengthened by histological staining of explanted human ventricular heart tissue. Alteration of three protein candidates from the proteomic screen and enhanced oxidative stress were confirmed in left ventricular cardiac tissue from a patient with PLN p. Arg14del end-stage HF.

What is the link between a mutation in PLN and a dysfunctional ER/mitochondria compartment? Often overlooked when studying cardiomyocyte function is that PLN not only regulates SERCA2a and $Ca^{2+}$ re-entry into the SR, but also importantly ascertains unperturbed $Ca^{2+}$ flux between the ER and the mitochondria. In principle, altered protein conformation of a dysfunctional mutant protein could directly affect localization and interaction of proteins crucial for ER/mitochondria integrity. Thus, it could, in a SERCA2-dependent or independent manner, alter $Ca^{2+}$ handling in a compartment critical for the functional integrity of ER-mitochondria interaction. Alternatively, haploinsufficiency and degradation of the mutant PLN protein could contribute to the disease phenotype. This hypothesis is supported by our data showing reduced PLN protein abundance by IF and proteomic analysis. An important focus of future studies is therefore to unequivocally determine whether the mutant PLN protein is expressed or immediately degraded by the UPR. Our data unveiled an unexpected beneficial action of $Ca^{2+}$ scavenging. Heterologous expression of $Ca^{2+}$-buffering proteins improved the PLN p. Arg14del disease phenotype as evidenced by an amelioration of functional, structural, and molecular abnormalities in PLN p. Arg14del EHTs.

In essence, the present study postulates PLN p. Arg14del-mediated dysfunction of the ER/mitochondria compartment as a novel molecular disease mechanism. This discovery opens new avenues for therapeutic intervention and target identification. The direct causal involvement of $Ca^{2+}$ irregularities and the beneficial effect of $Ca^{2+}$ scavenging are indicative of the potential for cytoplasmic $Ca^{2+}$ buffering as a promising therapeutic approach. This interconnects the pathology evoked by PLN p. Arg14del mutation with ongoing scientific discussions to optimize cytoplasmic $Ca^{2+}$ buffering as a treatment strategy in patients (Smith & Eisner, 2019).

# Materials and Methods

### Ethical approval

All human samples used in the present study were obtained after informed written consent by the patients. All experiments conformed to the principles set out in the WMA Declaration of Helsinki and the Department of Health and Human Services Belmont report (ethics agreement reference number NL30225.018.09; ethics commission Ruhr-Universität Bochum, reference number Reg.-Nr. 21.1/2013; University Utrecht protocol numbers: 12/387 and 15/252; Ethical Committee of the University Medical Center Hamburg-Eppendorf, reference number 532/116/9.7.1991).

### Reprogramming

A skin biopsy was taken from a PLN p. Arg14del mutation carrier diagnosed with familial DCM under local anesthesia after signature of informed consent (ethics agreement reference number NL30225.018.09). After washing in PBS, the skin piece was minced and placed in a T25 flask (Sarstedt) in fibroblast medium (DMEM with 10% FBS (PAA), 2 mM L-glutamine, and 0.5% penicillin and streptomycin (all Life Technologies)). Dermal fibroblasts growing out of the explants were cultured in a monolayer, expanded, and reprogrammed at passage 5 according to the previously published protocols (Takahashi et al, ,2007a, 2007b; Ohnuki et al, 2009). As described therein, transduction efficiency of the fibroblasts was improved by first introducing the mouse receptor for retroviruses Slc7a1 (encoded by pLenti6-Ubc-mSlc7a1). Subsequently, the human transcription factors OCT3/4, SOX2, KLF4, GLIS-1, and p53 shRNA were introduced with ecotropic retroviruses (encoded by pMXs-plasmids available from Addgene). All virus productions were performed in Plat-E packaging cells (Takahashi et al, 2007a) with retroviruses encoding the human transcription factors OCT3/4, SOX2, KLF4, GLIS-1, and p53 shRNA. Five days after transduction, dermal fibroblasts were harvested by trypsinization and replated at $8 \times 10^4$ cells/10 $cm^2$ on mitotically inactivated mouse embryonic fibroblasts (MEF strain CF-1). On the following day, the fibroblast medium was replaced by MEF-conditioned medium (supplemented with 10 ng/ml bFGF) followed by consecutive daily medium change. HiPSC clones were picked and transferred to Matrigel^TM-coated 48-well plates approximately 4 weeks after transduction and further expanded and cultured using the standard procedure (Breckwoldt et al, 2017).

### Karyotyping, mycoplasma

Karyotype and pluripotency analysis of hiPSC clones in master cell bank and mycoplasma screen of master cell bank and expanding hiPSCs was performed as recently described (Breckwoldt et al, 2017).

### Expansion and differentiation of human-induced pluripotent stem cells

Experiments were performed as recently described (Breckwoldt et al, 2017). In brief, hiPSCs were expanded from a master cell bank at passage 30–40 (PLN p. Arg14del) and passage 56–61 (PLNic) on Geltrex®-coated cell culture flasks in FTDA. Formation of embryoid bodies was performed in spinner flasks, and differentiation was conducted in Pluronic® F-127-coated cell culture flasks with a sequential administration of growth factor- and small molecule-based cocktails to induce mesodermal progenitors, cardiac progenitors, and cardiomyocytes. Dissociation of differentiated cardiomyocytes was performed with collagenase II (200 units/ml; Worthington, LS004176). Dissociated cardiomyocytes were analyzed for cardiac differentiation efficiency (cardiac troponin T) as recently described and subjected to EHT generation or 2D culture (Breckwoldt et al, 2017).

### Immunofluorescence staining of hiPSC

HiPSCs were rinsed briefly in 1x PBS and fixed for 15 min with 4% paraformaldehyde at room temperature. The hiPSCs were washed

three times with PBS for 5 min and incubated in blocking solution for 1 h at room temperature. Primary antibody incubation (Oct-4A or TRA-1-60(S) [Cell Signaling], 1:200 in antibody dilution buffer) was carried out overnight at 4°C in a humid chamber. The following day, the hiPSCs were washed three times with 1× PBS and incubated for 1 h at room temperature with the secondary antibody (Alexa Fluor 546 goat anti-rabbit IgG, 1:500 in antibody dilution buffer or Alexa Fluor 488 goat anti-mouse IgM μ chain, 1:500 in antibody dilution buffer, Molecular Probes Invitrogen) and DAPI for nuclear staining. After the final three PBS washing steps, the cells were mounted, analyzed, and stored at 4°C in the dark.

### Immunofluorescence of hiPSC-CM and microscopy

Prior to staining, hiPSC-CMs were fixed with 4% paraformaldehyde (PFA) in PBS for 10 min with two intermediate washing steps in 200 μl PBS/well and subsequently covered with parafilm and stored at 4°C in PBS until further processing.

For immunofluorescent staining of hiPSC-CM, PBS was removed and cells were permeabilized with either 0.2% Triton X-100 (Roth 3051.3) or 0.2% saponin (Sigma, S7900, from quillaja bark) in PBS for 5 min at room temperature (RT) followed by a washing step in 200 μl PBS/well for 5 min. Subsequently, non-specific binding sites were blocked with 5% Normal Goat Serum (NGS) for 20 min at RT. For the primary antibody incubation, cells were incubated overnight with 50 μl/well of the primary antibodies phospholamban (1:100, rabbit, Novus Biologicals, NBP2-19807), SERCA2 (1:100, mouse, Invitrogen, MA3-919), anti-8-hydroxy-2′-deoxyguanosine (8-OHdG, 1:50, mouse, Abcam, ab48508), GFP-FL (1:100, rabbit, Santa Cruz Biotechnology, sc-8334), calnexin (1:100, mouse, Novus Biologicals, AF18), VDAC1 (1:100, mouse, Abcam, ab14734), parvalbumin (1:100, rabbit, Abcam, ab181086) and α-actinin (1:500, mouse, Sigma, A7811) or α-actinin (1:500, rabbit, Sigma, A2543) diluted in antibody buffer (10 mM Tris, 155 mM NaCl, 2 mM EGTA, 2 mM MgCl$_2$, 1% (w/v) BSA, pH 7.5) at 4°C in a humid chamber under shaking. After three washing steps with 200 μl PBS/well, cells were incubated with 50 μl secondary antibody solution consisting of the secondary antibodies Alexa Fluor 647 goat anti-mouse (1:100, Invitrogen, A21236), Goat anti-rabbit DyLight® 550 (1:100 DyLight 550, Abcam, ab96884), and 4′,6-diamidino-2-phenylindole (DAPI, 1:100, Sigma, D9542) in a humid chamber on a shaker for 3h at RT. For staining of lipids, BODIPY (10 μM final concentration, D3911, Invitrogen) was added 30 min prior to the end of incubation with secondary antibodies. Thereafter, another five washing steps with 200 μl PBS/well were performed under agitation, and the 96-well plate was covered with parafilm and stored at 4°C until microscopy. The imaging system consisted of a Zeiss LSM 800 Airyscan confocal microscope, and the images were recorded using a 40× or 63× oil immersion objectives.

### In situ proximity ligation assay

Interactions between the ER and mitochondria were analyzed using an *in situ* proximity ligation assay (PLA). With this method, each distinct fluorescent spot represents an interaction between VDAC1 and IP$_3$R. Briefly, hiPSC-CMs were fixed with 4% PFA in PBS for 10 min at RT. After two intermediate washing steps in 200 μl PBS/well, the plates were subsequently covered with parafilm and stored at 4°C in PBS until further processing. For permeabilization, cells

were incubated in 100 μl 0.1% Triton X-100 PBS for 15 min at RT under agitation followed by one washing step in 200 μl PBS/well for 5 min. Non-specific binding sites were blocked for 30 min at 37°C in a humidity chamber with 100 μl of the blocking solution that was provided by the kit (Sigma, DUO2002). After removal of the blocking solution, the cells were incubated with the primary antibody combination VDAC (1:100, mouse, Abcam, ab14734) and IP$_3$R (1:500, rabbit, Abcam, ab5804) diluted in antibody buffer (10 mM Tris, 155 mM NaCl, 2 mM EGTA, 2 mM MgCl$_2$, 1% (w/v) BSA, pH 7.5) at 4°C in a humidity chamber on a shaker overnight. Negative controls were only incubated in antibody buffer without primary antibodies.

According to the manufacturer's instructions, the secondary antibodies conjugated either with the oligonucleotide PLA probe PLUS (Sigma, DUO92002) or with the oligonucleotide PLA probe MINUS (Sigma, DUO92004) were diluted 1:5 in the Duolink antibody diluent and allowed to incubate for 20 min at RT before usage. The primary antibody solutions were removed, and cells were washed twice in 200 μl 1× wash buffer A (Sigma, DUO82049) at RT under agitation followed by incubation with 40 μl/well of the PLA probe solution. The plates were then incubated in a pre-heated humidity chamber for 1 h at 37°C. The negative controls were incubated with either the PLA probe PLUS or the PLA probe MINUS.

Subsequently, the cells were washed twice with 200 μl wash buffer A for 5 min at RT followed by incubation for 30 min at 37°C in a pre-heated humidity chamber with a ligation solution (Sigma, DUO92007), consisting of two oligonucleotides and a ligase that hybridize the oligonucleotides to the two PLA probes if the distance between the targeted proteins is < 40 nm. The oligonucleotide of one of the PLA probes is then used as a primer for a rolling-circle amplification (RCA) enabling binding of fluorescently labeled oligonucleotides to the RCA product. For this amplification, cells were washed twice in 200 μl 1× wash buffer A for 5 min at RT and incubated in a pre-heated humidity chamber for 100 min at 37°C with 40 μl of an amplification solution provided by the kit (Sigma, DUO92007).

For counterstaining of the nuclei, the amplification solution was removed, and cells were washed twice in 200 μl 1× wash buffer B (Sigma, DUO82049) for 10 min at RT and for 1 min in 200 μl 1× wash buffer A and incubated with DAPI (1:100) in antibody buffer for 40 min at RT in a humidity chamber under agitation. Afterward, the cells were washed twice for 2 min with 1× wash buffer A and once with 200 μl 0.01× wash buffer B for 1 min and stored in PBS until microscopy. The imaging system consisted of a Zeiss LSM 800 Airyscan confocal microscope, and the images were recorded using a 40× oil immersion objective.

### Lentivirus production

To express the Ca$^{2+}$ sensor Fast-GCaMP6f-RS09 under control of the EF1a promoter together with a puromycin resistance, the lentiviral vector LeGO-EF1a i-Pur2 (derived from Addgene plasmid #27341 LeGO-iG2) was digested with BamHI and NotI. PCR was performed using Phusion polymerase and plasmid Fast-GCaMP6f-RS09 (Addgene 67160) as a template encoding GCaMP6f with an N-terminal 6× His Tag, a T7 Tag and an Xpress Tag with the following primer pair: 5′-GTCGTGAGGAATTCGGATCCaccATGGGTTCTCATC ATCATCATCATC and 5′- ATTTACGTAGCGGCCGCTTACTTCGCTG

TCATCATTTGTAC. PCR product and digested plasmid LeGO-EF1a i-Pur2 were incubated with 5X In-Fusion HD Enzyme Premix (In-Fusion HD Cloning Kit, Clontech) according to the recommendations of the manufacturer. Resulting clones were checked by restriction digest, PCR and were finally verified by sequencing. To express human parvalbumin (Homo sapiens parvalbumin (PVALB), transcript variant 2, mRNA; NCBI Reference Sequence: NM_001315532.2) under control of a ubiquitous EF1a promoter, a gene block was designed (Integrated DNA Technologies, IDT). This gene block comprised 5′-overhangs of 15 bp each homologous to the lentiviral plasmid LeGO-EF1a-iPur2 (kind gift from Boris Fehse, University Medical Center Hamburg-Eppendorf, Hamburg, Germany). After digestion of LeGO-EF1a-iPur2 with BamH I and Not I, both linearized plasmid and gene block were assembled using the In-Fusion Cloning Kit (Takara Clontech), generating LeGO-EF1a-huPVALB-iPur2. Resulting expression clones were checked by restriction digest, PCR and were finally verified by sequencing. Stocks of VSV-G pseudotyped viral particles were produced at the Vector Facility of the University Medical Center Eppendorf using lentiviral transfer plasmid LeGO-EF1a-huPVALB-iPur2 and packaging plasmids psPAX2 (Addgene plasmid #12260) and pMD2.G (Addgene plasmid #12259). LeGO-EF1a-iPur2 was employed to produce negative control lentivirus. After concentration by ultracentrifugation for 2 h at 4°C (140,000 $g$, SW32Ti rotor) on a 20% sucrose cushion, the pellet was resuspended in DPBS. The functional titer was determined by transduction of HEK293T and quantification by flow cytometry (FACSCantoII, BD Biosciences; FITC Channel).

## CRISPR/Cas9

PLN p. Arg14del mutation was verified in patient-derived hiPSCs by Sanger sequencing (Eurofins MWG Operon). For the generation of the isogenic controls, the CRISPR/Cas9 system derived from *Streptococcus pyogenes* (Sp) was applied. Identification of suitable PAM sequences and the design of the 20-nt long sgRNA were done with the online CRISPR design web tool provided by the Zhang laboratory (MIT, 2015; https://zlab.bio/guide-design-resources, CAACCATTGAAATGCCTCAA). The repair template was designed as a 103-nt long single stranded oligonucleotide (ssODN CTGCTGGTAT CATGGAGAAAGTCCAATACCTCACTCGCTCAGCTATAAGAAGAGCA TCAACCATTGAAATGCCTCAACAAGCACGTCAAAAGCTACAGAATCT) encoding the PLN WT sequence and carrying a silent mutation in the PAM sequence to avoid WT Cas9 cleavage of the ssODN. The ssODN was used in antisense direction as a 4 nmol Ultramer® DNA Oligo with standard desalting purification from Integrated DNA Technologies (IDT). The sgRNA was cloned into the pSpCas9(BB)-2A-GFP vector (Addgene, plasmid ID: 48138) according to the protocol published (Ran *et al*, 2013). For the ribonucleoprotein (RNP) approach, a 120 bp long sgDNA construct was designed encoding a T7 promoter preceding the sgRNA sequence. The sgRNA was generated by *in vitro* transcription with the High Scribe T7 quick HI yield kit (E2050, NEB) according to the manufacturer's instructions and cleaned up with the Ambio MegaClear kit (AM1903). The sgRNA was precipitated with ethanol and reconstituted in sterile PBS (stock concentration: 10 μg/μl). Recombinant GeneArtTM PlatinumTM Cas9 nuclease (Thermo Fisher Scientific/Invitrogen, B25641) was used.

PLN p. Arg14del hiPSCs were expanded in conditioned medium (murine embryonic fibroblasts) until 90–100% confluency. Two

hours before nucleofection, hiPSCs were incubated with Y-27632 (10 μM) and bFGF [30 ng]. HiPSCs were dissociated with Accutase® into single cells (5 min, 37°C). For nucleofection, 0.8x10⁶ hiPSCs were resuspended in 100 μl "P3 solution" (prepared by mixing 82 μl "Nucleofector solution" plus 18 μl supplement) according to the manufacturer's instructions (Lonza). For the DNA-based approach, 2 μg pSpCas9-sgRNA-2A-GFP plasmid and 5 μM ssODN repair template were mixed with the 100 μl cell suspension and transferred to the Amaxa nucleofection cuvette. For the RNP approach, 18 μg Cas9 was complexed with 4 μg sgRNA for 10 min at room temperature. Repair template (final concentration 5 μM) and the Cas9/sgRNA complex were mixed gently and the mixture was added to the 100 μl cell suspension into an Amaxa nucleofection cuvette. Nucleofection was performed with the 4D-Nucleofector™ (Lonza) according to the manufacturer's instructions. Nucleofection program CA-137 was applied. After nucleofection, the cells were incubated for 5 min in the cuvette at 37°C. HiPSC were plated on Matrigel™-coated 12-well culture dishes (0.8 × 10⁶ hiPSC per well) in conditioned medium supplemented with Y-27632 (10 μM). Medium was changed after 24 h. After 48 h, pSpCas9-sgRNA-2A-GFP-nucleofected hiPSCs were dissociated with Accutase® (5 min, 37°C), centrifuged (5 min, 200 ×$g$), and resuspended in PBS. GFP-positive cells were separated by FACS (drop rate of 1–2, nozzle of 100 μm, BD FACSAria™ Fusion) under sterile conditions, collected in conditioned medium with Y-27632 (10 μM), and plated in Matrigel-coated 6-well plates at 1.0–1.5 × 10³ cells per well. For the RNP approach, nucleofected hiPSCs without prior FASC sorting were plated at a density of 3.0 × 10³ cells/well. After 10–14 days of hiPSC expansion, individual clones were picked (10 μM Y-27632 1 h prior to picking) and transferred to 48- and subsequently 24-well dishes. Copy plates in the 24-well format were generated, and aliquots were frozen. DNA was isolated and sequenced to test the genome editing efficiency as well as to exclude modification of the 10 most probable off-target loci. Suitable isogenic control clones were chosen and sent for karyotyping. Both isogenic controls were used for the functional experiments in this study.

## Quantitative PCR

The reverse transcription of RNA to cDNA was performed with the "High Capacity cDNA Reverse Transcription Kit" (Applied Biosystems) according to the manufacturer's instructions. The qPCR was performed with SYBR Green (Thermo Fisher Scientific) in technical duplicates. The cardiac-specific housekeeping gene α-Actinin 2 (*ACTN2*) and glucuronidase-beta (*GUSB*) were used as reference transcripts for normalization. The target sequences were amplified within 40 cycles in an AbiPrism7900HT cycler (Applied Biosystems) according to the manufacturer's instructions. The following primer pairs were used: PLN (NM_002667.3), PLNwt/mut-for 5′-CAA TACCTCACTCGCTCAGC-3′, PLNwt/mut-rev: 5′-AGAGAAGCATCAC GATGATACAG-3′. PLNwt-for 5′-CTCGCTCAGCTATAAGAAGAGC-3′, PLNwt-rev: 5′-AGAGAAGCATCACGATGATACAG-3′. PLNmut-for 5′-CACTCGCTCAGCTATAAGAGC-3′, PLNmut-rev: 5′-AGAGAAGCATC ACGATGATACAG −3′, ACTN2 (NM_001103.3, NM_001278344.1, NM_001278343.1), ACTN2-for 5′-AAGGGGTGAAACTGGTGTCC-3′, ACTN2-rev: 5′-AGCAGACCTTCTTTGGCAGAT-3′. Primer pairs were adapted from Karakikes *et al*, (2015). Left ventricular heart samples

were collected after signature of informed consent (ethics commission Ruhr-Universität Bochum, reference number Reg.-Nr. 21.1/2013). Endoplasmic reticulum stress response genes analysed were recently described in Sicari *et al* (2020) (XBP1 splice variant -forward TGCTGAGTCCGCAGCAGGTG, -reverse: GCTGGCAGGCTCTGGGGA AG; CHOP -forward: CAGAACCAGCAGAGGTCACA, -reverse AGCT GTGCCACTTTCCTTTC; SCARA3 -forward: CGCTGCCAGAAGAACC TATC, -reverse AACCAGAGAGGCCAACACAG, BIP -forward: TGT TCAACCAATTATCAGCAAACTC, -reverse, TTCTGCTGTATCCTCTT CACCAGT; PDIA4 -forward: AGTGGGGAGGATGTCAATGC, -reverse TGGCTGGGATTTGATGACTG; HYOU1 -forward: GCAGACCTGTTGG CACTGAG, -reverse TCACGATCACCGGTGTTTTC). Fetal gene program PCR primers used were as follows: (MYH6 –forward: TCTTCTC CTCCTACGCAACTG, -reverse TTGAGATTTTCCCGGTGGAGAG; MYH7 –forward: GCTCTGTGTCTTTCCCTGCTGCTC, -reverse GCTCCTTC TCTGACTTGCGCAGG; COL1A1 –forward: GGGAATGCCTGGTGA ACGTG, -reverse CCTTGTCACCAGGGGCAC; NPPA –forward: GCT GCTTCGGGGGCAGGATG, -reverse TGCAGCAGAGACCCCAGGGG).

## Quantification of reactive oxygen/nitrogen species

Reactive oxygen and reactive nitrogen species (ROS/RNS) content was determined using the OxiSelect™ *in vitro* ROS/RNS Assay Kit (STA-347, Cell Biolabs, Inc., San Diego, CA) according to the manufacturer's instructions. For comparison of ROS/RNS in medium supernatant, EHTs were cultured for 24 h in 1.5 ml complete medium, of which 500 μl medium was centrifuged. Supernatant was snap-frozen in −80°C and thawed in a water bath at 37°C directly before the assay was conducted. Intracellular free radical content was measured after dissociating EHTs into single cells with papain and homogenization of $1 \times 10^6$ cells (pool of 4 EHTs, 48 technical, 8 biological replicates, three independent experiments) in 100 μl in PBS, and supernatant was snap-frozen after centrifugation and stored at −80°C until the assay was conducted. Fluorescence measurement was performed with a 485/538 nm filter set on a FluoroCount Plate Reader (Packard). Standard curves were constructed with dichlorofluorescin and hydrogen peroxide and prepared according to the manufacturer's instructions.

## Quantitative PCR mitochondria

Total and mitochondrial DNA was extracted using the DNeasy Blood & Tissue Kit (69506, Qiagen) following the manufacturer's protocol. Quantitative PCR was performed on the AbiPrism 7900HT Fast Real-Time PCR System (Applied Biosystems) utilizing SYBR Green/ROX qPCR Master Mix (K0222, Thermo Scientific). Primer sequences for mitochondrial DNA were as follows: mtND1 (Gene ID: 4535), mt-ND1for: 5′-ATGGCCAACCTCCTACTCCTCATT-3′, mt-ND1rev: 5′-TT ATGGCGTCAGCGAAGGGTTGTA-3′; mtND2 (Gene ID: 4536), mt-ND2for: 5′-CCATCTTTGCAGGCACACTCATCA-3′; mt-ND2rev: 5′-AT TATGGATGCGGTTGCTTGCGTG-3′; ACTB (NM_001101) (actin, beta) _for: 5′-CATGTACGTTGCTATCCAGGC-3′, ACTB_rev: 5′-CTCCTT AATGTCACGCACGAT-3′. Primer sequences as described (Ulmer *et al*, 2018).

## Analysis of mitochondrial respiration

The Seahorse™ XF96 extracellular flux analyzer was used to assess mitochondrial respiration as previously described (Mosqueira *et al*, 2019), using the Mito Stress Kit (Agilent Technologies). Briefly, cryopreserved isogenic sets of hiPSC-CMs were seeded into Matrigel™-coated (BD #356235) XF96-well plates at a density of approximately 5000 cells/mm$^2$. HiPSC-CMs were cultured for 2 days in RPMI1640 (US Biological Life Sciences #R9010-01) supplemented with B-27 with insulin (Life Technologies #0080085-SA), 2 mM L-glutamine (Life Technologies #25030-081), 10% fetal bovine serum (Gibco #16000044), and 0.6 mM CaCl$_2$. After 2 weeks, medium was exchanged for XF basal medium (Agilent Technologies #102353), supplemented with 10 mM glucose (Sigma #G7528), 1 mM sodium pyruvate (Sigma #S8636), and 2 mM L-glutamine (Life Technologies #25030-081) 1 h prior to conduction of the assay. Selective inhibitors were sequentially injected during the measurements (1.5 μM oligomycin, 0.4 μM FCCP, 1 μM rotenone; Agilent Technologies), following the manufacturer's instructions. The measured oxygen consumption rate (OCR) values were normalized to the number of cells in each well, quantified by 1:400 Hoechst 33342 staining (Sigma #B2261) in PBS (Gibco #14190-094) using fluorescence at 355 nm excitation and 460 nm emission in an automated imaging platform (CellaVista, Synentec). Statistical analysis was performed by using GraphPad Prism (v7, La Jolla, CA, USA) software, evaluated by unpaired one-way ANOVA test followed by Tukey post hoc test for correction of multiple comparisons, and differences were considered significant when *$P < 0.05$.

## Assessment of protein carbonylation

Detection of protein carbonylation in PLNic and PLN p. Arg14del hiPSC-CM was assessed using the Protein Carbonyl Assay Kit (Abcam #ab178020) according to the manufacturer's instructions. In brief, 2D hiPSC-CMs were washed with PBS and harvested in extraction buffer containing 50 mM DTT. Proteins were denatured by SDS and incubated with 2,4-dinitrophenylhydrazine (DNPH) or control derivatization solution for 15 min. The reaction was stopped by addition of neutralization buffer. Protein samples were separated by 10% SDS–PAGE, blotted onto nitrocellulose, and blocked in 10% non-fat milk in TBS-T. Membranes were probed with an anti-DNP antibody (1:5,000 in blocking buffer). Equal protein loading between DNPH and control-derivatized samples was ascertained by probing with an antibody against cMyBP-C. Proteins that underwent oxidation were detected by the anti-DNP antibody, only in the DNPH-derivatized samples.

## Western immunoblot analysis

For Western immunoblot analysis, EHTs were homogenized in 1x Mammalian Protein Extraction Reagent (M-PER; Thermo Fisher Scientific) supplemented with phosphatase inhibitor PhosSTOP (Sigma-Aldrich) and protease inhibitor cOmplete (Roche). Thereby, 70 μl of lysis buffer was used per EHT and 3–4 EHTs were pooled per sample. Homogenization was performed in a TissueLyser (Qiagen) using a metal bead. Samples were boiled at 95°C for 5 min. EHT homogenate samples were separated by SDS–PAGE (7.5–12%) and transferred to polyvinylidene difluoride (PVDF) or nitrocellulose membrane. After blocking non-specific binding sites with 10% (w/v) non-fat skimmed milk or 5% bovine serum albumin in 0.1% (v/v) Tween 20-TBS (Tris 20 mM, NaCl 137 mM; pH 7.6), membranes were incubated overnight at 4°C with primary antibodies (dilution

1:1,000–1:5,000), followed by horseradish peroxidase (HRP)-conjugated secondary antibody incubation on the next day (dilution 1:2,000). Specific protein bands were detected by enhanced chemi-luminescence (GE Healthcare) on a ChemiDoc (Bio-Rad) and quantified using the Gene Tools software (Syngene). Primary antibodies used were as follows: alpha-actinin (1:1,000, Sigma-Aldrich, A7811), phospholamban (1:1,000, Novus, NBP2-19807), phospho-lamban pSer16 (1:5,000, Badrilla, A010-12AP), pyruvate dehydrogenase (phospho S293) (Abcam, ab177461), pyruvate dehydrogenase (Abcam, ab168379), AMPK (phospho threonine 172) (Cell Signaling Technology, CST 2535), AMPKα (Cell Signaling Technology, CST D5A2), galectin-3 (Thermo Fisher, MA1-940), reticulocalbin-3 (Abcam, ab204178), LMCD1 (Thermo Fisher, MA5-25603), SAFB-like transcription modulator (Novus Biologicals, NBP2-38464), and calnexin (Novus Biologicals, AF18). Left ventricular heart samples were collected after informed consent (Ethical Committee of the University Medical Center Hamburg-Eppendorf, reference number 532/116/9.7.1991).

## Immunohistochemistry

The study met the criteria of the code of proper use of human tissue that is used in the Netherlands. The collection of the human heart tissue used for immunohistochemistry was approved by the advisory board of the biobank of the University Medical Center Utrecht, Utrecht, the Netherlands (protocol numbers: 12/387 and 15/252). Written informed consent was obtained or in certain cases waived by the ethics committee when obtaining informed consent was not possible due to death of the patient. For immunohistochemistry 3 PLN p. Arg14del explanted hearts, 3 explanted hearts with end-stage ischemic heart disease (remote area) and 3 healthy control hearts obtained during autopsy of patients that had died due to a non-cardiac cause were used. Sections were either pretreated with EDTA or boiled in citrate buffer. Primary antibodies used were as follows: PLN (Badrilla, A010-10AP), reticulocalbin-3 (Abcam, ab204178), LMCD1 (Atlas Antibodies, HPA024059), SAFB-like transcription modulator (Novus Biologicals, NBP2-38464), and anti-8-hydroxy-2′-deoxyguanosine (Abcam, ab48508). BrightVision Poly-AP-Anti mouse or rabbit IgG (Immunologic) was used as secondary antibody, and the signal was visualized with liquid permanent red (Dako). Quantification of 8-hydroxy-2′-deoxyguanosine (8-OHdG), reticulocalbin-3 (RCN3) aggregate-type formation was performed by two persons under blinded conditions.

## Transmission electron microscopy

EHTs were fixed in a mixture of 4% paraformaldehyde and 1% glutaraldehyde (Science Services, Germany) in 0.1 M phosphate buffer overnight at 4°C. Samples were rinsed three times in 0.1 M sodium cacodylate buffer (pH 7.2–7.4) and osmicated using 1% osmium tetroxide in cacodylate buffer. Following osmication, the samples were dehydrated using ascending ethanol concentrations, followed by two rinses in propylene oxide. Infiltration of the embedding medium was performed by immersion in a 1:1 mixture of propylene oxide and Epon (Science Services, Germany), followed by neat Epon and hardening at 60°C for 48 h. For light microscopy, semi-thin sections (0.5 μm) with longitudinal orientation were mounted on glass slides and stained for 1 min with 1% toluidine

blue. For electron microscopy, ultra-thin sections (60 nm) were cut and mounted on copper grids and stained using uranyl acetate and lead citrate. Sections were examined and photographed using an EM902 (Zeiss) electron microscope equipped with a TRS 2K digital camera (A. Tröndle, Moorenweis, Germany).

## EHT generation and maintenance, contractility recording

EHTs were generated from dissociated hiPSC-CM in a matrix from fibrinogen and thrombin as previously described, with $1 \times 10^6$ cells per EHT (Breckwoldt et al, 2017). EHTs were maintained in 24-well cell culture dishes in 1.5 ml EHT medium that was exchanged every second day. EHTs were subjected to force analysis between days 18 and 33 after casting. EHTs were fixed in formalin (4%) or snap-frozen in liquid nitrogen for DNA, RNA, or protein analysis. Contractility analysis in EHTs was performed by video-optical recording as described previously (Breckwoldt et al, 2017). Electrical pacing was performed with graphite pacing systems as recently described (Mannhardt et al, 2016). Contractility recording under serum free conditions was performed in modified Tyrode's solution (120 mM NaCl, 5.4 mM KCl, 1 mM MgCl$_2$, 0.1–5 mM CaCl$_2$, 0.4 mM NaH$_2$PO$_4$, 22.6 mM NaHCO$_3$, 5 mM glucose, 0.05 mM, Na$_2$EDTA, 25 mM HEPES) pre-equilibrated overnight (37 °C, 7% CO$_2$, 40% O$_2$, 53% N$_2$).

## Assessment of Ca$^{2+}$ transients in 2D hiPSC-CM

PLNic and PLN p. Arg14del hiPSC-CM were seeded onto gelatin-coated glass coverslips (24x24 mm, VWR) in a 6-well culture dish at a density of $2 \times 10^5$ hiPSC-CM per dish (10 cm$^2$). The hiPSC-CMs were cultured for $7 \pm 2$ days (RPMI, B-27) until analysis. 2.5 μM Fura 2-AM (5 mM stock in DMSO) was dissolved in 2 ml Tyrode's solution for 2D Ca$^{2+}$ transient analysis. The cardiomyocytes were washed with 2 ml Tyrode's solution and loaded with Fura 2-AM for 15 min protected from light. The Fura 2-AM-containing buffer was replaced by 3–4 ml Tyrode's solution for a 15-min washing step. The coverslips with the hiPSC-CM were removed from the culture wells for analysis with the IonOptix system (IonOptix, Milton, USA). The cells were continuously superfused with 37°C pre-warmed Tyrode's solution and paced at 0.5 Hz. Cells were included into the analysis, when they showed a regular beating pattern and followed the electrical pacing signal. Furthermore, cells should be either in a single or in a small cell cluster format (3–5 cells), but not part of larger cell aggregates. Initially, a stable baseline was recorded. Electrical stimulation was switched off, and 20 mM caffeine was applied to the cells. After 10–15 s, the perfusion was switched back to normal Tyrode's solution and cells were electrically stimulated again. The recording was continued for approximately 30 s. A maximum of three cells per coverslip was recorded.

## Proteomic analysis

EHTs from hiPSC-CM were dissociated with collagenase II solution (collagenase II, 200 units per ml), HBSS minus Ca$^{2+}$/Mg$^{2+}$, HEPES (10 mM), Y-27632 (10 μM), and BTS (30 μM, 4.5 h). Dissociated hiPSC-CMs were spun down (100 ×g, 5 min), supernatant was removed, and the pellet was frozen in liquid nitrogen /−80°C and subjected to proteome analysis. Samples were denatured by 6 M

urea and 2 M thiourea and reduced with 5mM DTT at 37°C for 1 h. Afterward, proteins were alkylated by 25 mM iodoacetamide and incubated in the dark for 1 h. Proteins were precipitated by adding 1 ml of ice-cold acetone and precipitated −20°C overnight. Samples were centrifuged at $16,000\times g$ at 4°C for 30 min, and the supernatant was discarded. The pellets were dried by SpeedVac (Thermo Fisher Scientific) for 10 min and resuspended in 0.1 M triethylammonium bicarbonate. Samples were digested with trypsin (protein: enzyme = 50:1) at 37°C overnight. Digestion was stopped by 1% trifluoroacetic acid, and the samples were purified on Bravo AssayMAP robot (Agilent) using C18 cartridges. Eluted peptides were SpeedVac dried and resuspended in 2% acetonitrile (ACN), 0.1% formic acid (FA) in LC-MS grade $H_2O$. Peptide samples were injected and separated by an UltiMate 3000 RSLCnano system (EASY-Spray column, 75 µm x 50 cm, 2 µm, Thermo Fisher Scientific) using the following LC gradient: 0–10 min: 4–10% B; 10–75 min: 10–30% B; 75–80 min: 30–40% B; 80–85 min: 40–99% B; 85–90 min: 99% B; 90–120 min: 4% B (A = 0.1% FA in $H_2O$, B = 80% ACN, 0.1% FA in $H_2O$). The separated peptides were directly injected into an Orbitrap Fusion Lumos Mass Spectrometer. Full MS spectra were collected using Orbitrap with scan range 350–1,500 $m/z$ and a resolution = 120,000. Most abundant ions from full MS scan were selected for MS2 with CID fragmentation in a linear ion trap, and cycle time was set to 3 s. Dynamic exclusion of 60 s and lock mass = 445.12003 $m/z$ were used.

Raw files were processed by Proteome Discoverer 2.2 (Thermo Fisher Scientific) and searched against UniProt/SwissProt human database (2018_05, 20349 protein entries) using Mascot 2.6.0. Trypsin was used for the enzymatic digest, and 2 missed cleavages were allowed. Carbamidomethylation on cysteines was selected as a fixed modification and oxidation on methionine as a variable modification. Precursor ion mass tolerance was set at 10 ppm and for fragment ion at 0.8 Da. Protein identification FDR confidence was set to high, and a minimum number of peptides per protein was 2. Precursor peak area was used for quantification and normalized to total peak area of each sample and scaled to average 100 across all samples. The mass spectrometry proteomics data have been deposited to the ProteomeXchange Consortium via the PRIDE (Perez-Riverol *et al*, 2019) partner repository with the dataset identifier PXD020175 and 10.6019/PXD020175.

Before applying enrichment and principal component analysis, data were further pre-processed with a more advanced processing and statistical analysis pipeline using the Ebayes algorithm of the limma package (Smyth). In specific, data were filtered for missing values. In specific, proteins with more than 30% missing values in all samples were filtered except the cases where there exist more than 90% missing values in one subphenotype and less than 10% missing values in the rest of the subphenotypes. In the latter case, zeros were imputed in the missing values of the subphenotype with more than 90% missing values. All remaining missing values were imputed with KNN-Impute method with k equal to 3 (default value). The relative quantities of the proteins were scaled using log2 transformation, and *P*-values were calculated and adjusted by Benjamini–Hochberg method. Volcano plots were generated with The R Package Ggplot2 (Wickham, 2009) showing as many significantly changing proteins id to avoid overlapping labels. Enrichment analysis has been conducted using David tool (Huang *et al*, 2009) This analysis included pathway terms KEGG (Kanehisa & Goto,

2000). Significant enriched terms were inferred with Benjamini–Hochberg-adjusted *P*-value threshold of 0.05.

To visualize the samples based on their proteomic profiles, we conducted principal component analysis (Jolliffe, 2011), projected the samples in the 3D space based on their 3 most significant principal components and colored samples based on their phenotype. The scikit-learn python library (Pedregosa *et al*, 2011) version 0.19.2 was used for the PCA, and Scree test (Cattell, 1966) was used to retain an adequate number of principal components to maintain at least the 90% variability of the dataset. The protein list was exported, and further calculations were performed in Excel. Co-expression networks were constructed using the Aracne-Ap method (Lachmann *et al*, 2016). The initial *P*-values were adjusted for multiple testing using Benjamini–Hochberg method (Ferreira & Zwinderman, 2006), and a threshold of 0.05 was used for the adjusted *P*-values to infer statistically significant changes.

## Transcriptomic analysis

### Library preparations and sequencing for RNA-seq

Four stages of differentiation, including undifferentiated hiPSC (day 0), mesodermal progenitors (day 3), cardiac progenitors (day 8), and hiPSC-CM (day 15), were analyzed. From these cells, RNA was extracted by using the Direct-zol RNA MicroPrep Kit (R2060) (Zymo Research). Libraries for stranded polyA+ RNA sequencing were prepared using 500 ng total RNA as input in the TruSeq Stranded mRNA Library Prep Kit (Illumina) following the instructions of the manufacturer. Paired-end sequencing was performed using Illumina NovaSeq 6000 System (2 × 50 bp) to a depth of 25 million reads per sample.

### Analysis of RNA-seq data

RNA-seq sequencing reads were trimmed for residual adapter sequences and low-quality 3′ ends. For mapping, we used STAR (v.2.6.1a, default parameters) (Dobin *et al*, 2013) to align all datasets to the human genome (hg38) and to quantify gene expression per sample, using gene annotations defined by Ensembl v.99. Read counts were normalized by sample by applying DESeq2's median of ratios (Love *et al*, 2014). We further used DESeq2 to identify differentially expressed genes between PLN p. Arg14del and PLNic at each stage of differentiation (*P*-adjusted < 0.01 and |Fold Change| ≥ 1.2). We identified enriched KEGG pathways (FDR of 5%) for the sets of differentially up- and downregulated genes by running gprofiler (Reimand *et al*, 2016). The generated RNA-seq datasets are available in the European Nucleotide Archive (ENA) repository under accession number PRJEB41838.

## Ca²⁺ transients in EHT

Sequential analysis of force and $Ca^{2+}$ transients was performed on a microscope stage using GCaMP6f as genetically encoded $Ca^{2+}$ sensor as recently reported (Saleem *et al*, 2020). In brief, hiPSC-CM EHTs were transduced by lentiviral particles (MOI of 0.3) containing the EF1a promoter—GCaMP6f RS09 plasmid (Addgene plasmid 67160) during casting. As fluorescent light source, a mercury lamp was connected to the Axiovert 200 M Zeiss microscope for excitation. A GFP excitation and emission filter set was used. Sequential analysis of force and $Ca^{2+}$ transients was performed in Tyrode's solution under electrical pacing with

### The paper explained

**Problem**

The mutation p.Arg14del in the gene encoding for human phospholamban (*hPLN*) causes dilated cardiomyopathy in patients. The precise molecular disease mechanisms remain incompletely understood. PLN is an important regulator of cytoplasmic calcium import into its internal stores, the sarcoplasmic (SR) and endoplasmic (ER) reticulum.

**Results**

Dermal fibroblasts from a PLN p.Arg14del mutation carrier were rederived to human-induced pluripotent stem cells (hiPSCs). The mutation was corrected by CRISPR/Cas9 technology. Control and PLN p.Arg14del hiPSC lines were differentiated into hiPSC cardiomyocytes (hiPSC-CMs) and analyzed in a three-dimensional engineered heart tissue organoid format. PLN p.Arg14del hiPSC-CM revealed significantly lower force development and the occurrence of an irregular beating pattern. Inter-organelle dysfunction between the ER and mitochondria was observed by RNA-seq and proteomic analyses and further validated by independent methodologies. Interestingly, cytoplasmic calcium lowering achieved by viral-mediated expression of calcium-binding proteins improved the cardiomyopathy phenotype.

**Impact**

We modeled PLN p.Arg14del cardiomyopathy in hiPSC-CM. This *in vitro* disease model allowed the demonstration of cytosolic calcium scavenging as a novel, promising therapeutic approach for future individualised precision medicine.

graphite electrodes in a 24-well culture format (Eppendorf, 030741005) on days 28–36 of EHT development.

### Statistical analysis

Data were expressed as means ± SEM. GraphPad Prism was used to compare between groups with Mann–Whitney *U*-test, unpaired, two-sided Student's *t*-test, and one-way or two-way ANOVA as indicated. Individual hiPSC-CM EHTs or single hiPSC-CM from 9 different differentiation batches from two different clones were considered as biological replicates. An overview of exact *P*-values and number of biological replicates for each experiment is placed in the manuscript as Appendix Fig S10. Statistical details.

## Data availability

Relative quantities of all proteins for the proteomics data of the paper are provided in Dataset EV1. Code availability: Not applicable. Proteomics data are deposited as a PRIDE partner repository with the dataset identifier PXD020175 (http://www.ebi.ac.uk/pride/archive/projects/PXD020175). The RNA-seq datasets are available in the European Nucleotide Archive (ENA) repository under accession number PRJEB41838.

**Expanded View** for this article is available online.

## Acknowledgements

We would like to acknowledge Prof. Hendrik Milting for contributing RNA samples of DCM patients carrying a PLN p. Arg14del mutation. We greatly appreciate the assistance of Kristin Hartman (UKE mouse pathology core facility) and UKE FACS Core unit and the team approach of hiPSC and CRISPR/Cas9 group at IEPT/UKE. We are grateful to expert technical assistance provided by Emanuela Szpotowicz. This study was supported by the British Heart Foundation RM/13/30157, European Research Council (ERC-AG IndivuHeart), Deutsche Forschungsgemeinschaft (DFG Es 88/12-1, DFG HA 3423/5-1, DFG CU 53/5-1), the Werner-Otto-Stiftung (7/92 to FC), the Deutsche Stiftung für Herzforschung (F/19/19 to FC), the British National Centre for the Replacement Refinement & Reduction of Animals in Research (NC3Rs CRACK-IT grant 35911-259146), the German Ministry of Education and Research (BMBF) and the Centre for Cardiovascular Research (DZHK), and the Freie und Hansestadt Hamburg. M. Mayr is a British Heart Foundation (BHF) Chair Holder (CH/16/3/32406) with BHF program grant support (RG/16/14/32397) and is part of the Marie Skłodowska-Curie Innovative Training Network TRAIN-HEART (http://train-heart.eu), and JR was supported by a fellowship of the Studienstiftung des deutschen Volkes. Open Access funding enabled and organized by Projekt DEAL.

## Author contributions

Reprogramming (SL, IB), CRISPR/Cas (AEK, IB, SL, DM, CD), PCR (AEK), cardiac differentiation (AEK, BK, TS), EHT generation and maintenance (AEK, BK, TS, US), force measurement (AEK, BK, TS), Ca$^{2+}$ transient 2D (FF), immunofluorescence/Duolink of 2D hiPSC-CM (AEK, JR), proteomics/bioinformatics (MM, KT, XY), RNA sequencing (JR-O, GP, NH), IHC (PvdK, AV), ROS quantification (BMU), Mt DNA-, quantitative PCR (ML, AP), TEM (MS), Seahorse (DM, CD), WB (FC, BK, AK), GCaMP6f (US), patient samples (YP), concept (FC, CD, SH, TE, AH), funding (SH, TE, AH), writing (FC, TE, AH, AK).

## Conflict of interest

T.E. and A.H. are co-founders of EHT Technologies, Hamburg.

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
