## [Review Process File · EMBO Molecular Medicine]

Impairment of the ER/mitochondria compartment in human cardiomyocytes with PLN p.Arg14del mutation

Friederike Cuello, Anika Eike Knaust, Umber Saleem, Malte Loos, Janice Raabe, Diogo Mosqueira, Sandra Laufer, Michaela Schweizer, Petra van der Kraak, Frederik Flenner, Bärbel Ulmer, Ingke Braren, Xiaoke Yin, Konstantinos Theofilatos, Jorge Ruiz-Orera, Giannino Patone, Birgit Klampe, Thomas Schulze, Angelika Piasecki, Yigal Pinto, Aryan Vink, Norbert Hübner, Sian Harding, Manuel Mayr, Chris Denning, Thomas Eschenhagen, and Arne Hansen

DOI: 10.15252/emmm.202013074

Corresponding author: Arne Hansen (ar.hansen@uke.de)

Review Timeline:

Submission Date:	7th Jul 20
Editorial Decision:	21st Jul 20
Revision Received:	9th Mar 21
Editorial Decision:	19th Mar 21
Revision Received:	31st Mar 21
Accepted:	7th Apr 21

Editor: Lise Roth

Transaction Report:

21st Jul 2020

Dear Prof. Hansen,

Thank you for submitting your work to EMBO Molecular Medicine. We have now heard back from the three referees who agreed to evaluate your manuscript. As you will see below, the reviewers raise substantial concerns on your work, which unfortunately preclude its publication in EMBO Molecular Medicine in its current form.

The reviewers find that the question addressed by the study is of potential interest, however, they remain unconvinced that some of the major conclusions are sufficiently supported by the data. In particular, given the limited novelty of the findings, providing evidence of causality between the PLN Arg14Del mutation and the ER-stress/mitochondrial pathway, as well as investigating the molecular mechanism will be a prerequisite for publication.

Addressing the reviewers' concerns in full will be necessary for further considering the manuscript in our journal. Still, revising the manuscript according to the referees' recommendations appears to require a lot of additional work and experimentation. We are therefore ready to extend the deadline to 6 months with the understanding that acceptance of the manuscript would entail a second round of review. EMBO Molecular Medicine encourages a single round of revision only and therefore, acceptance or rejection of the manuscript will depend on the completeness of your responses included in the next, final version of the manuscript. For this reason, and to save you from any frustrations in the end, I would strongly advise against returning an incomplete revision.

When submitting your revised manuscript, please carefully review the instructions that follow below. Failure to include requested items will delay the evaluation of your revision:

- 1) A .docx formatted version of the manuscript text (including legends for main figures, EV figures and tables). Please make sure that the changes are highlighted to be clearly visible.
- 2) Individual production quality figure files as .eps, .tif, .jpg (one file per figure).
- 3) A .docx formatted letter INCLUDING the reviewers' reports and your detailed point-by-point responses to their comments. As part of the EMBO Press transparent editorial process, the point-by-point response is part of the Review Process File (RPF), which will be published alongside your paper.
- 4) A complete author checklist, which you can download from our author guidelines (<https://www.embopress.org/page/journal/17574684/authorguide#submissionofrevisions>). Please insert information in the checklist that is also reflected in the manuscript. The completed author checklist will also be part of the RPF.
- 5) Please note that all corresponding authors are required to supply an ORCID ID for their name upon submission of a revised manuscript.

6) Before submitting your revision, primary datasets produced in this study need to be deposited in an appropriate public database (see <https://www.embopress.org/page/journal/17574684/authorguide#dataavailability>). Please remember to provide a reviewer password if the datasets are not yet public. The accession numbers and database should be listed in a formal "Data Availability " section (placed after Materials & Method). Please note that the Data Availability Section is restricted to new primary data that are part of this study.

7) We would also encourage you to include the source data for figure panels that show essential data. Numerical data should be provided as individual .xls or .csv files (including a tab describing the data). For blots or microscopy, uncropped images should be submitted (using a zip archive if multiple images need to be supplied for one panel). Additional information on source data and instruction on how to label the files are available at .

8) Our journal encourages inclusion of *data citations in the reference list* to directly cite datasets that were re-used and obtained from public databases. Data citations in the article text are distinct from normal bibliographical citations and should directly link to the database records from which the data can be accessed. In the main text, data citations are formatted as follows: "Data ref: Smith et al, 2001" or "Data ref: NCBI Sequence Read Archive PRJNA342805, 2017". In the Reference list, data citations must be labeled with "[DATASET]". A data reference must provide the database name, accession number/identifiers and a resolvable link to the landing page from which the data can be accessed at the end of the reference. Further instructions are available at .

9) We replaced Supplementary Information with Expanded View (EV) Figures and Tables that are collapsible/expandable online. A maximum of 5 EV Figures can be typeset. EV Figures should be cited as 'Figure EV1, Figure EV2" etc... in the text and their respective legends should be included in the main text after the legends of regular figures.

- Additional Tables/Datasets should be labeled and referred to as Table EV1, Dataset EV1, etc. Legends have to be provided in a separate tab in case of .xls files. Alternatively, the legend can be supplied as a separate text file (README) and zipped together with the Table/Dataset file. See detailed instructions here: .

10) The paper explained: EMBO Molecular Medicine articles are accompanied by a summary of the articles to emphasize the major findings in the paper and their medical implications for the non-specialist reader. Please provide a draft summary of your article highlighting

11) For more information: There is space at the end of each article to list relevant web links for further consultation by our readers. Could you identify some relevant ones and provide such information as well? Some examples are patient associations, relevant databases, OMIM/proteins/genes links, author's websites, etc...

12) Every published paper now includes a 'Synopsis' to further enhance discoverability. Synopses are displayed on the journal webpage and are freely accessible to all readers. They include a short stand first (maximum of 300 characters, including space) as well as 2-5 one-sentences bullet points that summarizes the paper. Please write the bullet points to summarize the key NEW findings. They should be designed to be complementary to the abstract - i.e. not repeat the same text. We encourage inclusion of key acronyms and quantitative information (maximum of 30 words / bullet point). Please use the passive voice. Please attach these in a separate file or send them by email, we will incorporate them accordingly.

Please also suggest a striking image or visual abstract to illustrate your article. If you do please provide a png file 550 px-wide x 400-px high.

13) As part of the EMBO Publications transparent editorial process initiative (see our Editorial at <http://embomolmed.embopress.org/content/2/9/329>), EMBO Molecular Medicine will publish online a Review Process File (RPF) to accompany accepted manuscripts.

In the event of acceptance, this file will be published in conjunction with your paper and will include the anonymous referee reports, your point-by-point response and all pertinent correspondence relating to the manuscript. Let us know whether you agree with the publication of the RPF and as here, if you want to remove or not any figures from it prior to publication.

EMBO Molecular Medicine has a "scooping protection" policy, whereby similar findings that are published by others during review or revision are not a criterion for rejection. Should you decide to submit a revised version, I do ask that you get in touch after six months if you have not completed it, to update us on the status.

I look forward to receiving your revised manuscript.

Yours sincerely,

Lise Roth

Lise Roth, PhD
Editor
EMBO Molecular Medicine

Photos 400-800 DPI

*Additional important information regarding figures and illustrations can be found at <http://bit.ly/EMBOPressFigurePreparationGuideline>

***** Reviewer's comments *****

Referee #1 (Remarks for Author):

The manuscript entitled "Impairment of rough ER/mitochondria compartment in human cardiomyocytes with PLN p.Arg14Del mutation" by Knaust et al. describes a new mechanism by which the Arg14Del heterozygous mutation in phospholamban (PLN) causes dilated cardiomyopathy (DCM). For this, skin fibroblasts isolated from a patient carrying the PLN Arg14Del mutation were reprogrammed into a hiPSC line and differentiated into cardiomyocytes using a 3D engineered heart tissue. An isogenic control hiPSC line was created using CRISPR/Cas9 and human hearts were used to correlate some of the findings from hiPSC-CM. Calcium measurements, proteomic analysis and electron microscopy revealed irregular beatings, lower force and prolonged Ca²⁺ transient decay time in mutant hiPSC-CM. These changes were associated with reduced expression of ER and mitochondrial proteins, dilation of the ER with presence of large lipid droplets, indicative of a severe impairment of the rough ER. Results showed a reversal of the phenotype after transduction of a Ca²⁺ scavenging protein which has important ramifications for potential therapeutic treatment. The present study is well-designed and used advanced technology to create two independent hiPSC-CM clones validated by karyotyping and sequencing to exclude off-target effects, which were then differentiated in 3D heart tissue. Unbiased proteomic was used to identify major proteins and pathways enriched or defective in hiPSC-CM with the Arg14Del mutation. Mitochondrial function was assessed using the Seahorse analyzer. PLN is a key protein that controls Ca²⁺ re-uptake by the SR and this mutation accounts for a significant number of DCM in the Dutch population. Overall, this study is important because it defines a novel mechanism causing PLN Arg14Del-associated DCM in humans implicating abnormalities of the rough ER and mitochondria. Some concerns were identified mostly with the different sub-cellular localization of PLN in hiPSC-CM with the Arg14Del mutation, which is not consistent with previous reports

documenting the formation of large aggregates with perinuclear staining of PLN. Some of the data lacking statistical analysis are over-stated, and in the case of GCaMP6f delivered using lentivirus, a proper control is missing. These issues should be addressed and are summarized below.

Major concerns

1. The immunofluorescence data (Figure 1e) documents lower PLN signal in PLN Arg14Del hiPSC-CM compared to control hiPSC-CM. The images show as expected, a majority of perinuclear staining of PLN in PLNic cells. However, PLN is exclusively nuclear in hiPSC-CM with the Arg14Del mutation, suggesting that the Arg14Del mutation disrupts the sub-localization of residual PLN. This is not consistent with a previous report (Karakikes et al. Nat. Com. 2015) where PLN Arg14Del was expressed in one side of the cytoplasm in hiPSC-CM derived from a patient with the PLN Arg14Del mutation. Haghghi and colleagues also showed that PLN Arg14Del has ER and perinuclear staining when expressed in Hek293 cells (Haghghi et al., PNAS 2006). How do the authors reconcile this discrepancy? The mutation may disrupt interaction with proteins critical for ER and mitochondrial function due to the mislocalization of PLN itself, which is interesting. This should be discussed at a minimum if not investigated further.

2. The conclusion that only the monomeric form of PLN is reduced in PLN Arg14Del compared to PLNic is not supported by the Western blot data (Figure 2A). The pentameric form of PLN appears reduced in PLN Arg14Del cells at least from the Western blot provided, which should be consistent with the quantitative analysis (Figure 2B).

3. The conclusion of the experiment with isoprenaline is that it "significantly increases PLN Ser16 phosphorylation of monomeric and pentameric PLN in PLNic and that a similar effect is observed in PLN Arg14Del cells". Do the authors mean that ISO has no effect on the monomeric and pentameric forms of PLN in PLNArg14Del cells? This should be clarified. Furthermore, the quantitative analysis (Figure 2D) does not support the conclusion because there is no significant difference shown in Figure 2D. Also, total PLN should be shown by Western blot and used to calculate the ratio of total PLN/phosphoPLN.

4. The sample number used in Figure 2B and D should be increased to reach the conclusion that Arg14Del does not affect the neighboring Ser16 PKA phosphorylation site.

5. Figure 7E: total PDH and AMPK should be shown, and the quantitation should include total/phosphorylated form.

6. One characteristic of the PLNArg14Del mutation is the formation of large aggregates in the perinuclear region of mutant patient hearts that have been reported before. Data from the study indeed show the presence of perinuclear aggregates for PLN and 8-OHdG in Arg14Del mutant hearts. Can 8-OHdG aggregates be observed in hiPSC-CM with the PLM Arg14Del mutation?

7. The PLNArg14Del mutation causes DCM. Up-regulation of fetal cardiac genes would be expected in hiPSC-CM with the PLM Arg14Del mutation. Is-it the case?

8. The rescue experiment using a lentivirus expressing GCaMP6f is interesting. However, there is no indication that a control lentivirus was used in PLNic and PLN Arg14Del cells. The proper control should be included. Also, what is the effect of GCaMP6f on PLN protein expression and sub-cellular localization? Is-there a rescue of PLN expression after GCaMP6f delivery?

Minor issue

9. Figure 1e legend: part of the legend is missing and indicates RNA and PCR which is probably a mistake considering that the figure shows immunofluorescence data.

Referee #2 (Comments on Novelty/Model System for Author):

The authors developed an interesting and useful tool by deriving p.Arg14del iPSCs. The initial characterization of these cells, and of derived cardiomyocytes, is well performed and convincing. These cells will be a great resource for future studies. However, this is only the beginning of the story, and the paper does not provide new mechanical insights to understand the origin of the cardiac phenotype developing in p.Arg14del patients. More work is needed to identify the implicated molecular mechanisms.

Referee #2 (Remarks for Author):

The paper by Knaust et al. aims at characterizing the p.Arg14del mutation in phospholamban, causing cardiomyocyte dysfunction and ultimately heart failure in humans. The mutation is dominant because heterozygous carriers are affected by ventricular dysfunction and cardiac arrhythmia. Mouse models exist but do not completely recapitulate the human disease. The authors used therefore another approach and produced induced pluripotent stem cells (iPSCs) from an affected individual. An appropriate control was created by correcting the mutation in iPSCs using CRISPR/Cas9 (herein PLNic). Thus, iPSC-derived cardiomyocytes, obtained either by using a 2D-culture system or via producing engineered heart tissues (EHTs), demonstrated the expected functional defect, as observed in p.Arg14del patients.

Then, the authors evaluated a series of parameters allowing for the evaluation of calcium handling and force production, as well as arrhythmia. In summary, p.Arg14del cardiomyocytes demonstrated normal calcium response and sarcoplasmic reticulum function but developed lower contractile force. The authors used therefore a proteomic approach to search for candidate proteins responsible for the observed phenotype. A number of proteins (>1500) were differentially expressed in p.Arg14del cardiomyocytes. Among those were proteins defining several key biological processes, in particular mitochondrial activity and dynamic, and structure of the rough endoplasmic reticulum (ER). The authors conclude that impairment of microdomains important for mitochondria-ER interaction play important role in the development of p.Arg14del-induced heart failure.

Globally, this study is well performed. The methods are appropriate and the results are consistent. Therefore, the authors proposed a thorough characterization of p.Arg14del iPSC-derived cardiomyocytes, mostly pharmacological. Nevertheless, the results are not completely novel. A number of papers have been published, describing the p.Arg14del mutation and their downstream detrimental effects in cardiomyocytes. The present study confirms nicely previous findings. However, if the idea was to go one step further and to identify the molecular mechanisms responsible for the development of the disease, this is not completely accomplished. In particular, the proteomic analysis is interesting but remains purely descriptive. Predictions drawn from this analysis should be experimentally tested.

Main points

1. A weakness in the paper consists in the fact that the characterization is performed only using

differentiate cardiomyocytes, the end-product of the differentiation process. In order to study the development of the disease, the authors should consider evaluating cellular intermediates during the production of iPSC-derived cardiomyocytes (using an established factor-induced 2D culture system). A transcriptomic analysis at different time points following induction of differentiation should be particularly informative. A comparison with publicly available datasets should also be useful.

2. One point that is not evaluated is the importance of the wild-type allele in the development of the phenotype in p.Arg14del iPSCs. Work in the mouse suggests that the wild-type protein plays a role in the disease process. The authors should use CRISPR/Cas9 to invalidate the wild-type allele in p.Arg14del iPSCs, and assess functionally these cells expressing solely p.Arg14del phospholamban to evaluate the role of the wild-type protein.

3. In the same vein, homozygote human carriers demonstrate an exacerbated cardiac phenotype. In a reciprocal experiment, it would be interesting to produce iPSCs carrying the p.Arg14del mutation on both alleles.

4. From their proteomic analysis, the authors conclude that mitochondrial dysfunction plays an important role in establishing the diseased phenotype in p.Arg14del iPSCs. However, the causal relationship is not formally demonstrated. Mitochondrial dysfunction could well be a consequence of upstream functional defects. The differences in the mitochondrial compartment between the two genotypes, as described in Figures 6 and 7, and in Supplementary Figure 3, 4 and 5, are rather small, and might not account for the full phenotype observed in p.Arg14del iPSC-derived cardiomyocytes.

5. The authors should probably investigate the phenotype developing in the rough ER further. This likely results from the accumulation of misfolded proteins, p.Arg14del phospholamban, at the origin of cardiomyocyte dysfunction more than downstream mitochondrial dysregulation.

6. Finally, the correction of phenotype by GCaMP6f is not explained. The efficacy of a calcium buffer in a situation described as normal in terms of calcium handling and SR function is not easy to understand. This should be investigated in more details, using other calcium modulators, to confirm the available data.

Referee #3 (Remarks for Author):

These authors generated PLN p.Arg14del iPSC-derived CMs and EHTs. They studied the contractile and calcium phenotypes as well as the morphological and (in part) functional phenotypes of ER and mitochondria. Finally, they use a calcium-based intervention to rescue the lack-of-force phenotype of mutant cells. The study is scientifically sound and well conducted. However, the novelty is limited (e.g. PMID: 25923014) and lack of mechanistic findings undermine the enthusiasms for this paper. It is unclear if the claimed ER/mitochondria alterations are causative of disease or just associated with it. Finally, no mechanisms underlying the effect of PLN mutation on ER/PLN structure and function are investigated.

Specific comments.

- It is unclear to me, in what the p.Arg14del mutation in PLN will functionally result. Will the mutation be causing a truncated protein more susceptible to degradation (hence, the reduction in PLN levels)?

- Figure 1E. It looks like that other than PLN, also SERCA2a protein was less presented in PLN mutant CMs. Is this something expected? Does reduction in PLN affect SERCA2a levels as well.

- Figure 2A. I am not convinced that mutated CMs show less abundance in PLN monomeric form compared to isogenic controls due to the large variability of band intensities (protein abundance) among samples. Please provide clearer blots. In addition, why the PM form appears exclusively in the NB samples? Looking at these it looks to me that also the PM form is reduced in the p.Arg14del mutant.

- Figure 2C. Why the authors did not show here the total PLN levels? It is hard to appreciate the conserved phosphorylation levels in the mutant vs. control without the total PLN levels (which should be reduced in the mutant).

- The observations about mitochondrial dysfunction and ER enlargement (despite in absence of canonical ER stress - PERK, IRE1a, ATF6 unchanged) in mutant CMs are interesting, but quite associative. It is unclear if the mutant PLN affects directly (or indirectly) mitochondria number and function as well as ER function. Where the ER-mitochondria contacts disrupted? Does mutant PLN preferentially interact with ER/SR related protein? It is difficult to determine if the observed alterations in PLN mutant CMs where the cause or the consequences of contractile dysfunction.

- If the authors hypothesized that calcium alterations do not entirely explain the phenotype of CMs mutants, why they chose a calcium-related intervention to rescue the phenotype? It would be of interest if mitochondrial or ER interventions might exert an effect in this sense.

Manuscript EMM-2020-13074**Response to the reviewers**

We would like to thank the editor and reviewers for their constructive criticism on our manuscript entitled: **Impairment of the ER/mitochondria compartment in human cardiomyocytes with PLN p.Arg14del mutation**. We have performed extensive additional experiments to address the reviewers' concerns in full and have revised the manuscript accordingly. We also have reworked on the introduction and discussion to increase clarity and comprehensiveness. New data are highlighted in yellow in the text. Please find as follows a detailed point-by-point response to all comments.

Referee #1 (Remarks for Author):

The manuscript entitled "Impairment of rough ER/mitochondria compartment in human cardiomyocytes with PLN p.Arg14Del mutation" by Knaust et al. describes a new mechanism by which the Arg14Del heterozygous mutation in phospholamban (PLN) causes dilated cardiomyopathy (DCM). For this, skin fibroblasts isolated from a patient carrying the PLN Arg14Del mutation were reprogrammed into a hiPSC line and differentiated into cardiomyocytes using a 3D engineered heart tissue. An isogenic control hiPSC line was created using CRISPR/Cas9 and human hearts were used to correlate some of the findings from hiPSC-CM. Calcium measurements, proteomic analysis and electron microscopy revealed irregular beatings, lower force and prolonged Ca²⁺ transient decay time in mutant hiPSC-CM. These changes were associated with reduced expression of ER and mitochondrial proteins, dilation of the ER with presence of large lipid droplets, indicative of a severe impairment of the rough ER. Results showed a reversal of the phenotype after transduction of a Ca²⁺ scavenging protein which has important ramifications for potential therapeutic treatment. The present study is well-designed and used advanced technology to create two independent hiPSC-CM clones validated by karyotyping and sequencing to exclude off-target effects, which were then differentiated in 3D heart tissue. Unbiased proteomic was used to identify major proteins and pathways enriched or defective in hiPSC-CM with the Arg14Del mutation. Mitochondrial function was assessed using the Seahorse analyzer. PLN is a key protein that controls Ca²⁺ re-uptake by the SR and this mutation accounts for a significant number of DCM in the Dutch population. Overall, this study is important because it defines a novel mechanism causing PLN Arg14Del-associated DCM in humans implicating abnormalities of the rough ER and mitochondria. Some concerns were identified mostly with the different sub-cellular localization of PLN in hiPSC-CM with the Arg14Del mutation, which is not consistent with previous reports documenting the formation of large aggregates with perinuclear staining of PLN. Some of the data lacking statistical analysis are over-stated, and in the case of GCaMP6f delivered using lentivirus, a proper control is missing. These issues should be addressed and are summarized below.

We thank the reviewer for his/her kind comments and have addressed the concerns in our manuscript.

Major concerns

#1.1. The immunofluorescence data (Figure 1e) documents lower PLN signal in PLN Arg14Del hiPSC-CM compared to control hiPSC-CM. The images show as expected, a majority of

perinuclear staining of PLN in PLNic cells. However, PLN is exclusively nuclear in hiPSC-CM with the Arg14Del mutation, suggesting that the Arg14Del mutation disrupts the sub-localization of residual PLN. This is not consistent with a previous report (Karakikes et al. Nat. Com. 2015) where PLN Arg14Del was expressed in one side of the cytoplasm in hiPSC-CM derived from a patient with the PLN Arg14Del mutation. Haghghi and colleagues also showed that PLN Arg14Del has ER and perinuclear staining when expressed in Hek293 cells (Haghghi et al., PNAS 2006). How do the authors reconcile this discrepancy? The mutation may disrupt interaction with proteins critical for ER and mitochondrial function due to the mislocalization of PLN itself, which is interesting. This should be discussed at a minimum if not investigated further.

The reviewer raises an important point. The IF image of the PLN and SERCA2 staining was indeed not representative of the localization of the proteins. We have performed extensive additional IF experiments, with a representative image now incorporated into the manuscript as a new **Figure 1E**. Here, we show perinuclear, ER and cytoplasmic staining for PLN and SERCA2. Additional images in Appendix **Figure S1G, H** demonstrate PLN and SERCA2 localization in reference to the established marker of the Z-disc, α -actinin. The data do not indicate a consistent difference in localization of PLN between the PLNic and PLN p.Arg14del hiPSC-CM lines, but an overall weaker staining intensity for PLN in PLN p.Arg.14del, suggesting lower PLN protein abundance in disease. We described the new results in the manuscript in **line 116-119**.

#1.2. The conclusion that only the monomeric form of PLN is reduced in PLN Arg14Del compared to PLNic is not supported by the Western blot data (Figure 2A). The pentameric form of PLN appears reduced in PLN Arg14Del cells at least from the Western blot provided, which should be consistent with the quantitative analysis (Figure 2B).

We increased the replicate number for the PLN western blot analysis in **Figure 2A, B** to n=8. The data revealed significantly lower band intensities for the monomeric PLN band (MM) for PLNic and PLN p.Arg14del compared to left-ventricular non-failing human heart tissue (NFH), but no difference in the PLN pentameric (PM) band intensities. Between PLNic and PLN p.Arg14del, no difference was observed for both, MM and PM, PLN bands. We have incorporated the new data to generate a new **Figure 2A, B**. We described the new results in the manuscript in **line 119-121**.

#1.3. The conclusion of the experiment with isoprenaline is that it "significantly increases PLN Ser16 phosphorylation of monomeric and pentameric PLN in PLNic and that a similar effect is observed in PLN Arg14Del cells". Do the authors mean that ISO has no effect on the monomeric and pentameric forms of PLN in PLNArg14Del cells? This should be clarified. Furthermore, the quantitative analysis (Figure 2D) does not support the conclusion because there is no significant difference shown in Figure 2D. Also, total PLN should be shown by Western blot and used to calculate the ratio of total PLN/phosphoPLN.

We increased the number of replicates to n=8 and performed statistical analysis by two-way ANOVA. This showed a significant increase in the band intensity corresponding to Ser16 phosphorylation of pentameric (PM) PLN after exposure to isoprenaline (ISO) for PLNic. No significant increase in S16 phosphorylation of monomeric (MM) PLN after ISO treatment was detectable for PLNic. This might be attributable to low signal intensity and thus

corresponding high scatter of signals, which made the quantification of band intensities more variable. For PLN p.Arg14del, neither S16 phosphorylation of the PM nor the MM PLN reached statistical significance. We suspect that this was due to similar issues as described for PLNic. The results are reflected by the new representative blot incorporated in **Figure 2C** and the according scatter plot that summarizes the results for all n-numbers (**Figure 2D**). Additionally, we included in this point-by-point response to the reviewer a scatter plot that reflects the data of PLN S16 phosphorylation normalized to total PLN expression (below right panel, n=5). For consistency of our data analysis that throughout normalized to α -actinin expression, we would prefer not to show this in the original manuscript (please see also comment to Referee #3.4.), but would of course adhere to the reviewers' preference. To emphasize the effect of ISO on both genotypes (statistical analysis shown in **Figure 2E, F**), we also included average contraction peaks at baseline and after ISO exposure as shown in **Appendix Figure S2A**. We described the new results in the manuscript in **line 121-125 and 126-130**.

#1.4. The sample number used in Figure 2B and D should be increased to reach the conclusion that Arg14Del does not affect the neighboring Ser16 PKA phosphorylation site.

We increased replicate numbers to n=8 and consequently replaced the representative blots shown in **Figure 2A, B, C, D**. We described the new results in the manuscript in **line 119-121 and 121-125**.

#1.5. Figure 7E: total PDH and AMPK should be shown, and the quantitation should include total/phosphorylated form.

We would like to thank the reviewer for this important point. We increased replicate numbers to n=9. For presentation and analyses, we were guided by the reference paper of Cardenas et al, Cell 2010 (PMID: 20655468), who showed representative western blots for phosphorylated and total PDH as well as phosphorylated and total AMPK and used the band intensities for the phosphorylated proteins for quantification and analyses.

The western immunoblots (**Figure 8 A, B, C, D**) showed significantly higher band intensities for pPDH at Ser293 and pAMPK at Thr172 in PLN p.Arg14del compared to PLNic at baseline. We chose to analyze the phosphorylation status of PDH and AMPK as a surrogate to visualize the potentially beneficial effect of Ca^{2+} scavenging in response to heterologous GCaMP6f and parvalbumin expression. For PLN p.Arg14del, phosphorylation of AMPK was significantly reduced after expression of GCaMP6f and parvalbumin compared to the respective PLN p.Arg14del control group. Phosphorylation of PDH was also significantly reduced in the

presence of parvalbumin. The new data are incorporated into the manuscript as a new **Figure 8 A, B, C, D**) and the new results described in the manuscript in **line 283-286**.

#1.6. One characteristic of the PLNArg14Del mutation is the formation of large aggregates in the perinuclear region of mutant patient hearts that have been reported before. Data from the study indeed show the presence of perinuclear aggregates for PLN and 8-OHdG in Arg14Del mutant hearts. Can 8-OHdG aggregates be observed in hiPSC-CM with the PLM Arg14Del mutation?

We thank the reviewer for pointing out this important aspect. We performed additional experiments to support our observation that oxidative stress is a confounding factor for PLN p.Arg14del cardiomyopathy. Using immunofluorescence, we demonstrated an enhanced perinuclear aggregate-type staining pattern for 8-hydroxy desoxyguanosine (8-OHdG), a bona fide marker for oxidative stress, in PLN p.Arg14del 2D hiPSC-CM compared to PLNic. A representative IF-image is now incorporated as a new **Figure 6F** in the manuscript. We paralleled the IF experiments by western immunoblot analysis performed with an anti-2,4-dinitrophenylhydrazine antibody that detected the presence of carbonyl groups modified by 2,4-dinitrophenylhydrazine as a surrogate of irreversibly oxidized proteins. This revealed increased carbonylation of proteins in PLN p.Arg14del compared to PLNic corroborating the findings obtained with IF analysis and 8-OHdG staining. The new data are now incorporated into the manuscript as a new **Figure 6G**. We described the new results in the manuscript in **line 219-223**.

#1.7. The PLNArg14Del mutation causes DCM. Up-regulation of fetal cardiac genes would be expected in in hiPSC-CM with the PLM Arg14Del mutation. Is-it the case?

We analyzed EHTs from both genotypes for the upregulation of prototypical fetal marker genes and proteins by RT-qPCR, RNAseq and proteomic analyses (**Appendix Figures S9, S4, S5**,). We could not detect significant differences regarding the expression of fetal marker genes between the genotypes, which might be related to the maturation state of hiPSC-CM and represent a general difficulty to induce hypertrophy in hiPSC-CM models and therefore reflect a limitation. This information has been included into the revised manuscript in the results in **line 240-241 and 293-296**.

#1.8. The rescue experiment using a lentivirus expressing GCaMP6f is interesting. However, there is no indication that a control lentivirus was used in PLNic and PLN Arg14Del cells. The proper control should be included. Also, what is the effect of GCaMP6f on PLN protein expression and sub-cellular localization? Is-there a rescue of PLN expression after GCaMP6f delivery?

This is a very important question and we thank the reviewer for raising this issue. We have performed extensive additional experiments to address this. In this regard, we have repeated all experiments by including an empty lentivirus control group. Additionally, we also included in our experiments a group per genotype that was transduced with the Ca²⁺-binding EF-hand protein parvalbumin to corroborate that the beneficial action of GCaMP6f overexpression is indeed causally mediated by Ca²⁺ scavenging. With all groups, we performed western immunoblot analysis (**Appendix Figure S8**) and IF analysis (**Appendix Figure S7**) for the detection of PLN and SERCA protein levels and localization. This revealed no difference in PLN protein abundance and similar perinuclear localization between the

genotypes in response to viral transduction-mediated scavenging of Ca^{2+} . By various alternate readouts including assessment of PDH and AMPK phosphorylation levels (**Figure 8**), contractile function (**Figure 8**), ER stress response (**Figure 8**), proximity of ER and mitochondrial proteins by Proximity Ligation assays (**Figure EV2**) and TEM (**Figure EV3**), we demonstrated improvement of ER/mitochondria structure, response to stress and contractile function in response to cytoplasmic Ca^{2+} scavenging. This information has been included into the revised manuscript in the results **in line 276-314**.

Minor issue

#1.9. Figure 1e legend: part of the legend is missing and indicates RNA and PCR which is probably a mistake considering that the figure shows immunofluorescence data.

We revised this in the new version of the manuscript.

Referee #2 (Comments on Novelty/Model System for Author):

The authors developed an interesting and useful tool by deriving p.Arg14del iPSCs. The initial characterization of these cells, and of derived cardiomyocytes, is well performed and convincing. These cells will be a great resource for future studies. However, this is only the beginning of the story, and the paper does not provide new mechanical insights to understand the origin of the cardiac phenotype developing in p.Arg14del patients. More work is needed to identify the implicated molecular mechanisms.

Referee #2 (Remarks for Author):

The paper by Knaust et al. aims at characterizing the p.Arg14del mutation in phospholamban, causing cardiomyocyte dysfunction and ultimately heart failure in humans. The mutation is dominant because heterozygous carriers are affected by ventricular dysfunction and cardiac arrhythmia. Mouse models exist but do not completely recapitulate the human disease. The authors used therefore another approach and produced induced pluripotent stem cells (iPSCs) from an affected individual. An appropriate control was created by correcting the mutation in iPSCs using CRISPR/Cas9 (herein PLNic). Thus, iPSC-derived cardiomyocytes, obtained either by using a 2D-culture system or via producing engineered heart tissues (EHTs), demonstrated the expected functional defect, as observed in p.Arg14del patients.

Then, the authors evaluated a series of parameters allowing for the evaluation of calcium handling and force production, as well as arrhythmia. In summary, p.Arg14del cardiomyocytes demonstrated normal calcium response and sarcoplasmic reticulum function but developed lower contractile force. The authors used therefore a proteomic approach to search for candidate proteins responsible for the observed phenotype. A number of proteins (>1500) were differentially expressed in p.Arg14del cardiomyocytes. Among those were proteins defining several key biological processes, in particular mitochondrial activity and dynamic, and structure of the rough endoplasmic reticulum (ER). The authors conclude that impairment of microdomains important for mitochondria-ER interaction play important role in the development of p.Arg14del-induced heart failure.

Globally, this study is well performed. The methods are appropriate and the results are consistent. Therefore, the authors proposed a thorough characterization of p.Arg14del iPSC-

derived cardiomyocytes, mostly pharmacological. Nevertheless, the results are not completely novel. A number of papers have been published, describing the p.Arg14del mutation and their downstream detrimental effects in cardiomyocytes. The present study confirms nicely previous findings. However, if the idea was to go one step further and to identify the molecular mechanisms responsible for the development of the disease, this is not completely accomplished. In particular, the proteomic analysis is interesting but remains purely descriptive. Predictions drawn from this analysis should be experimentally tested.

We thank the reviewer for his/her kind comments and have addressed the concerns in our manuscript.

Main points

#2.1. A weakness in the paper consists in the fact that the characterization is performed only using differentiated cardiomyocytes, the end-product of the differentiation process. In order to study the development of the disease, the authors should consider evaluating cellular intermediates during the production of iPSC-derived cardiomyocytes (using an established factor-induced 2D culture system). A transcriptomic analysis at different time points following induction of differentiation should be particularly informative. A comparison with publicly available datasets should also be useful.

We would like to thank the reviewer for his/her constructive comments and for raising this important point. As suggested, we have performed RNA sequencing analysis of PLN^{ic} and PLN p.Arg14del hiPSC during the cardiac differentiation procedure at crucial time points and stages at day 0 (hiPSCs), at day 3 (mesoderm progenitors), at day 8 (cardiac progenitors) and day 15 (early cardiomyocytes). The principal component analysis revealed that the samples from both genotypes clustered according to the stage of the differentiation procedure (**Appendix Figure S4C**). This is also reflected by the pattern of the differentially expressed stage-specific marker genes presented as a heatmap (**Appendix Figure S4A, B**). Notably, the comparison of the transcriptional profile in samples at day 15 revealed lower abundance of cardiomyocyte- and endothelial-specific markers in PLN^{p.Arg14del}. This difference suggested a delayed rather than reduced cardiac differentiation in PLN^{p.Arg14del} since proteomic analysis performed at a later stage of cardiac maturation in a 3D culture format did not show significant differences in the abundance of cardiomyocyte/myofilament marker proteins (**Table EV1**). The transcriptional profile at day 15 also revealed differential expression of a number of ER/mitochondria contact site marker genes (e.g. *CANX*, *MFN2*, *VDAC1-3*, *ITPR2*) as shown by the transcriptional profile of individual selected genes (**Appendix Figure S5**). Notably, fetal gene program markers were not different in PLN^{ic} versus PLN p.Arg14del and only a subset of ER/UPR stress response markers were elevated in PLN p.Arg14del. The new information has been included into the revised manuscript in a new **Appendix Figure S4A, B** and the relevant information has been included into the revised manuscript in the results in **line 228-241**.

#2.2. One point that is not evaluated is the importance of the wild-type allele in the development of the phenotype in p.Arg14del iPSCs. Work in the mouse suggests that the wild-type protein plays a role in the disease process. The authors should use CRISPR/Cas9 to invalidate the wild-type allele in p.Arg14del iPSCs, and assess functionally these cells expressing solely p.Arg14del phospholamban to evaluate the role of the wild-type protein.

Please find our comment below at #2.3.

#2.3. In the same vein, homozygote human carriers demonstrate an exacerbated cardiac phenotype. In a reciprocal experiment, it would be interesting to produce iPSCs carrying the p.Arg14del mutation on both alleles.

We agree with the reviewer that these are very interesting additional aspects that deserve clarification. However, we would like to point out that addressing this issue represents a new scientific project by itself and lies beyond the time and experimental scope of the present study. Prerequisite is a study like the present one, which described the molecular phenotype of the heterozygous mutation as it is present in human mutant carriers that develop DCM. We acknowledge the importance to understand the relative contributions of the wildtype and mutant (p.Arg14del) PLN protein for the disease phenotype. Quantitative PCR (**Figure 1C, D**) with wildtype- and mutant-specific primer pairs suggested co-expression of both mRNAs. Unfortunately, proteomic analysis after trypsin digestion of the proteins did not allow to discriminate between both PLN forms and genotype-specific PLN antibodies capable to differentiate between wildtype and mutant protein do not exist yet. Several attempts to isolate and characterize native PLN from hiPSC-CM homogenates from both genotypes failed, but will be the focus of intense future experimental efforts.

#2.4. From their proteomic analysis, the authors conclude that mitochondrial dysfunction plays an important role in establishing the diseased phenotype in p.Arg14del iPSCs. However, the causal relationship is not formally demonstrated. Mitochondrial dysfunction could well be a consequence of upstream functional defects. The differences in the mitochondrial compartment between the two genotypes, as described in Figures 6 and 7, and in Supplementary Figure 3, 4 and 5, are rather small, and might not account for the full phenotype observed in p.Arg14del iPSC-derived cardiomyocytes.

We thank the reviewer for raising this important point. Alterations in the mitochondrial compartment are demonstrated in the present study by independent methodology: quantification of mitochondrial DNA (abundance; **Appendix Figure S3B**), differential abundance of mitochondrial proteins by proteomic analysis (**Figure 5B, Table EV1**), altered mitochondrial morphology with associated lipid droplets (TEM; **Figure 5D, E**; IF for BODIPY, **Figure 5F**), basal/maximal respiration and ATP production (function; **Figure 6A-D**). These data concordantly suggest impaired mitochondrial function in PLN p.Arg14del compared to PLNic. Importantly, these observations were paralleled by alterations of the ER (proteomic analysis; **Figure 5B; Table EV1**, TEM, **Figure 5D, E**, IF of 2D hiPSC-CM, **Figure EV1**, Proximity Ligation Assays (**Figure EV2**)). The phosphorylation status of PDH and AMPK have been shown previously to directly indicate alterations of the ER/mitochondria interface (PMID: 20655468). Accordingly, we show higher phosphorylation of PDH and AMPK in PLN p.Arg14del (**Figure 8A-D**), enhanced oxidative stress (**Figure 6**) and partial reversal of the PLN p.Arg14del disease phenotype upon heterologous expression of Ca²⁺ scavenging proteins GCaMP6f and parvalbumin (**Figure 8**). In aggregate, we demonstrate an evidence obtained from independent methodologies that PLN p.Arg14del evokes an impairment of the ER/mitochondria interface that is less pronounced in the corresponding isogenic control or after Ca²⁺ scavenging. The new information has been included as new or updated figures into the revised manuscript.

#2.5. The authors should probably investigate the phenotype developing in the rough ER further. This likely results from the accumulation of misfolded proteins, p.Arg14del phospholamban, at the origin of cardiomyocyte dysfunction more than downstream mitochondrial dysregulation.

We thank the reviewer for this important comment. We agree that the precise molecular disease mechanisms and the sequence of maladaptive events has not been fully unraveled in the present study, but remains the topic of intense investigation in future studies. This is also related to the inability to determine whether PLN p.Arg14del protein is expressed (please also see, comment to **#2.3**). Nevertheless, the following results unequivocally point to the ER as the primary affected compartment of the disease:

- The role of PLN in the regulation of ER and SR Ca^{2+} homeostasis via SERCA2 is well established (PMID: 23223628)
- We provide evidence in the present study that SR function is unperturbed. We show that ER structure and function is altered in PLN p.Arg14del including the ER/mitochondria contact sites, reflected by the altered phosphorylation status of PDH and AMPK and this was associated with an accumulation of lipid droplets and oxidative stress.
- Inter-organelle communication between the ER and the mitochondria via ER/mitochondria contact sites is well established (PMID: 20655468)
- We provide evidence for alterations in mitochondrial number, shape and function. The ER/mitochondria contact sites in PLN p.Arg14del showed increased sensitivity to the SERCA2 inhibitor thapsigargin as shown by Proximity Ligation Assays (PLA). Stronger phosphorylation of PDH in PLN p.Arg14del provides further evidence for a disrupted ER/ mitochondrial contact site. Attempts to directly measure the distance between ER structures and mitochondria by TEM did not succeed in EHTs.
- Ca^{2+} scavenging by GCaMP6f improved contractile force in PLN p.Arg14del hiPSC-CM, was accompanied by improved ER morphology, paralleled by a reduction in the PDH and AMPK phosphorylation state and a reduction in ER stress response markers for both Ca^{2+} scavenging interventions.

The accumulation of misfolded proteins likely contributes to the pathological ER phenotype, which is compatible with the signal for ER stress response as detected by quantitative PCR (Figure 8F-H) and the prominent presence of aggresomes. Furthermore, the amelioration of the disease phenotype in the presence of Ca^{2+} -scavenging proteins implied that Ca^{2+} irregularities contribute to this scenario.

#2.6. Finally, the correction of phenotype by GCaMP6f is not explained. The efficacy of a calcium buffer in a situation described as normal in terms of calcium handling and SR function is not easy to understand. This should be investigated in more details, using other calcium modulators, to confirm the available data.

We would like to thank the reviewer for this suggestion. We generated a novel lentiviral vector encoding for the Ca^{2+} binding EF hand protein parvalbumin and a suitable empty lentiviral control vector. We performed additional experiments using hiPSC-CM of both genotypes either control transduced or after transduction with the lentivirus encoding for GCaMP6f or parvalbumin. In 2D hiPSC-CM, heterologous transduction with GCaMP6f or

parvalbumin did not result in any overt alteration in PLN or SERCA protein expression as shown by IF and western immunoblot analysis (**Appendix Figures S7, S8**). Proximity ligation assays (PLA) performed with the combination of an anti-VDAC1 (localizing to the outer mitochondrial membrane) and an IP3R-antibody (localizing to the ER) revealed no difference in the PLN^{ic} between the control, GCaMP6f or parvalbumin group (**Figure EV2**). Thapsigargin exposure (1 μ M, 5 hour incubation) was applied as an established ER stress protocol (PMID: 21266244). A small increase in signal intensity was observed in the PLN^{ic} hiPSC-CM. In sharp contrast were the results obtained for PLN p.Arg14del hiPSC-CM. In comparison to the control transduced group, PLA fluorescence intensity was higher after expression of the Ca²⁺ binding proteins GCaMP6f and parvalbumin. Interestingly, thapsigargin exposure strongly decreased PLA fluorescence in all PLN p.Arg14del groups, suggesting lower tolerance to ER stress compared to PLN^{ic}. In the 3D EHT format, transduction with the lentivirus encoding for GCaMP6f led to higher force development in the PLN p.Arg14del group. Phosphorylation of PDH and AMPK was higher for PLN p.Arg14del versus PLN^{ic} under control virus condition. Functional improvement by Ca²⁺ scavenging was accompanied by a reduction in the phosphorylation state of AMPK and PDH for PLN p.Arg14del (**Figure 8A-D**). No changes in the transcription of fetal gene program markers was detectable between genotypes and interventions (**Appendix Figure 9**). A subset of ER stress response markers were slightly elevated in PLN p.Arg14del and reduced in the presence of Ca²⁺ scavengers (**Figure 8**). In aggregate, the data indicate an important role of the ER/mitochondrial compartment for the PLN p.Arg14del cardiomyopathy and the impact of cytoplasmic Ca²⁺.

Referee #3 (Remarks for Author):

These authors generated PLN p.Arg14del iPS-derived CMs and EHTs. They studied the contractile and calcium phenotypes as well as the morphological and (in part) functional phenotypes of ER and mitochondria. Finally, they use a calcium-based intervention to rescue the lack-of-force phenotype of mutant cells. The study is scientifically sound and well conducted. However, the novelty is limited (e.g. PMID: 25923014) and lack of mechanistic findings undermine the enthusiasms for this paper. It is unclear if the claimed ER/mitochondria alterations are causative of disease or just associated with it. Finally, no mechanisms underlying the effect of PLN mutation on ER/PLN structure and function are investigated.

We thank the reviewer for his/her kind comments and have addressed the concerns in our manuscript.

Specific comments.

#3.1. It is unclear to me, in what the p.Arg14del mutation in PLN will functionally result. Will the mutation be causing a truncated protein more susceptible to degradation (hence, the reduction in PLN levels)?

We would like to thank the reviewer for the constructive and positive feedback on our manuscript and would like to point out that this manuscript aimed to reach beyond the description of aggregate formation as a causal disease contributor. In fact, we provided a detailed molecular characterization of the ER and mitochondrial compartment, supported by unbiased transcriptomic and proteomic analysis. Ultimately, the disease contribution by

perturbed Ca²⁺ cycling evidenced by the Ca²⁺ scavenging rescue experiments emphasized that cytoplasmic Ca²⁺ is an important culprit of the disease.

Providing an answer to the overarching question how the PLN p.Arg14del mutation exactly leads to the disease phenotype is important and relevant. As also eluded to in the response to Referee #2.2. and #2.3., quantitative PCR (**Figure 1C, D**) with wildtype- and mutant-specific primers suggested co-expression of both mRNAs. The lower PLN protein abundance detected by proteomic analysis (**Appendix Figure S3D**), while no difference was detectable by western immunoblot analysis (**Figure 2B; Appendix Figure S8A**), could suggest that the mutant form might be subject to degradation. Unfortunately, proteomic analysis did not allow us to discriminate between wildtype and mutant PLN versions and genotype-specific PLN antibodies do not exist yet. Several attempts to isolate and characterize native PLN from hiPSC-CM homogenates of both phenotypes failed. In aggregate, we cannot provide evidence or determine if mutant PLN protein is expressed in PLN p.Arg14del hiPSC-CM.

#3.2. Figure 1E. It looks like that other than PLN, also SERCA2a protein was less presented in PLN-mutant CMs. Is this something expected? Does reduction in PLN affects SERCA2a levels as well.

We agree that this is important information to add. As PLN p.Arg14del leads to the expression of a mutant PLN protein, a lower overall PLN protein abundance was expected. However, reduced PLN protein abundance did not impact on SERCA protein expression. We now included a more representative IF image for PLN and SERCA localization (**Figure 1E**) and a representative western immunoblot demonstrating similar protein abundance for both, PLN and SERCA2 (**Appendix Figure S8A-D**), as well as proteomic analysis showing reduced abundance for PLN and no difference for SERCA2 (**Appendix Figure S3D, E**).

#3.3. Figure 2A. I am not convinced that mutated CMs show less abundance in PLN monomeric form compared to isogenic controls due to the large variability of band intensities (protein abundance) among samples. Please provide clearer blots. In addition, why the PM form appears exclusively in the NB samples? Looking at these it looks to me that also the PM form is reduced in the p.Arg14del mutant.

We increased replicate numbers to n=8 for the western blot experiments, which we have now incorporated into the manuscript in a new **Figure 2A, B**. The data showed lower abundance of monomeric PLN in both, PLN^{ic} and PLN p.Arg14del versus non-failing human heart tissue (NFH). Between PLN^{ic} and PLN p.Arg14del, no difference was observed for both, MM and PM PLN bands. The reviewer raised the question whether the band corresponding to pentameric PLN was detectable only in NFH samples. We apologize for this misunderstanding. PLN PM band was detected for all three samples only under non-boiled conditions and not under boiled conditions. No difference in pentameric PLN abundance was detected between all three samples under non-boiled conditions. We updated the manuscript accordingly.

#3.4. Figure 2C. Why the authors did not show here the total PLN levels? It is hard to appreciate the conserved phosphorylation levels in the mutant vs. control without the total PLN levels (which should be reduced in the mutant).

We increased the replicate number for the western immunoblots in Figure 2C, D to n=8. For the additional replicates, we also normalized the PLN pSer16 band intensities to PLN total (n=5). The graph is shown below on the right in comparison to the normalization to α -actinin (on the left). Since the results for the normalization of phosphorylation of PLN at Ser16 to α -actinin versus normalization to total PLN were not different, we preferred to only show the normalization to α -actinin in the manuscript for better consistency with our results and the ability to include higher replicate numbers. We hope the reviewer concurs, but of course we would adapt to the reviewers' preference.

#3.5. The observations about mitochondrial dysfunction and ER enlargement (despite in absence of canonical ER stress - PERK, IRE1a, ATF6 unchanged) in mutant CMs are interesting, but quite associative. It is unclear if the mutant PLN affects directly (or indirectly) mitochondria number and function as well as ER function. Where the ER-mitochondria contacts disrupted? Does mutant PLN preferentially interact with ER/SR related protein? It is difficult to determine if the observed alterations in PLN mutant CMs where the cause or the consequences of contractile dysfunction.

We thank the reviewer for this important comment. We agree that the precise molecular disease mechanisms and the sequence of maladaptive events has not been fully unraveled in the present study, but remains the topic of intense investigation in future studies. This is also related to the inability to determine if PLN p.Arg14del protein is expressed (please also see, comment to **#2.3**). Nevertheless, the following indications unequivocally point to the ER as the primary affected compartment:

- The role of PLN in the regulation of ER and SR Ca^{2+} homeostasis via SERCA2 is well established (PMID: 23223628)
- We provide evidence in the present study that SR function is unperturbed. We show that ER structure and function is altered in PLN p.Arg14del including the ER/mitochondria contact sites, reflected by the altered phosphorylation status of PDH and AMPK and this was associated with an accumulation of lipid droplets and oxidative stress.
- Inter-organelle communication between the ER and the mitochondria via ER/mitochondria contact sites is well established (PMID: 20655468)
- We provide evidence for alterations in mitochondrial number, shape and function. The ER/mitochondria contact sites in PLN p.Arg14del showed increased sensitivity to the SERCA2 inhibitor thapsigargin as shown by Proximity Ligation Assays (PLA). Stronger phosphorylation of PDH in PLN p.Arg14del provides further evidence for a disrupted ER/ mitochondria contact site. Attempts to directly measure the distance between ER structures and mitochondria by TEM did not succeed in EHTs.

- Ca^{2+} scavenging by GCaMP6f improved contractile force in PLN p.Arg14del hiPSC-CM, was accompanied by improved ER morphology, paralleled by a reduction in the PDH and AMPK phosphorylation state and a reduction in ER stress response markers for both Ca^{2+} scavenging interventions.

The accumulation of misfolded proteins likely contributes to the pathological ER phenotype, which is compatible with the signal for ER stress response as detected by quantitative PCR (Figure 8F-H) and the prominent presence of aggresomes. Furthermore, the amelioration of the disease phenotype in the presence of Ca^{2+} -scavenging proteins implied that Ca^{2+} irregularities contribute to this scenario.

The reviewer raised the question whether mutant PLN might preferentially interact with ER/SR related proteins. We agree that it is an important question. If it was the case that PLN p.Arg14del would exert altered affinity for SERCA2a versus the housekeeping SERCA2b, it would be relevant for ER Ca^{2+} homeostasis in non-myocyte cells. Specifically, since differences in the Ca^{2+} -affinity and -turnover between SERCA2a and SERCA2b (PMID: 11679415, PMID: 12663488) have been reported previously. However, we would like to point out the difficulties to distinguish morphologically and functionally between ER and SR (PMID: 23223628, PMID: 21071716). A clear visual separation between the ER and SR compartment has not been described yet. Previous reports suggested that PLN affinity for both isoforms might be comparable (PMID: 1326945), studies that investigate this for PLN p.Arg14del have not been published to our knowledge.

#3.6. If the authors hypothesized that calcium alterations do not entirely explain the phenotype of CMs mutants, why they chose a calcium-related intervention to rescue the phenotype? It would be of interest if mitochondrial or ER interventions might exert an effect in this sense.

Given the role of PLN in the regulation of intracellular Ca^{2+} handling, combined with previous reports showing Ca^{2+} irregularities in PLN p.Arg14del, we analyzed cytoplasmic Ca^{2+} transients in 2D cultures after Fura-2 loading (**Figure 4A-E; Appendix Figure S2B-F**) and in EHTs after heterologous expression of the genetically encoded Ca^{2+} sensor GCaMP6f (PMID: 31956082). As chance-finding, we discovered improved force development of PLN p.Arg14del in the presence of GCaMP6f, the effects described in detail in **Appendix Figure S6** and validated this with the additional dataset that included experiments performed with another Ca^{2+} binding EF-hand protein parvalbumin as shown in **Figure 8**. As described in our comment #2.5 and #3.5, our data indicate that the ER is the primarily affected compartment in PLN p.Arg14del cardiomyopathy. Accordingly, we investigated ER inhibition by thapsigargin as an established ER stress protocol and detected altered interaction between VDAC1 and IP₃R by Proximity Ligation Assays in PLN p.Arg14del hiPSC-CM (**Figure EV2**).

19th Mar 2021

Dear Prof. Hansen,

Thank you for the submission of your revised manuscript to EMBO Molecular Medicine. We have now received the enclosed reports from the three referees who re-reviewed your manuscript. As you will see, they are supportive of publication, and I am therefore pleased to inform you that we will be able to accept your manuscript, once the following editorial points will be addressed:

1) Main manuscript text:

- Please answer/correct the changes suggested by our data editors in the main manuscript file (in track changes mode). This file will be sent to you in the next couple of days. Please use this file for any further modification.
- Please remove the highlights in the text.
- The abstract should be a single paragraph (without subsections) not exceeding 175 words. Please modify accordingly.
- Please replace the heading "Online Methods Supplement/Experimental procedures" by "Material and Methods".
- Material and Methods:
 - o Reprogramming: We note that you refer to already published methods, please make sure that enough information is nevertheless provided to ensure reproducibility of the experiments.
 - o Human samples: please include a statement confirming that written informed consent was obtained from all subjects and that the experiments conformed to the principles set out in the WMA Declaration of Helsinki and the Department of Health and Human Services Belmont report.
 - o Assessment of protein carbonylation: please provide the antibody dilution.
- Please provide independent sections for "Statistics" and "Data availability section".
- Statistics: Please also indicate in the figures or in the legends the exact $n=$ and exact $p=$ values along with the statistical test used. You may provide these values as a supplemental table in the Appendix file.
- "Data availability section": the primary datasets produced in this study need to be deposited in an appropriate public database and the accession numbers added to this section. Please also note that the datasets have to be made public before acceptance of the manuscript.
- Please replace the heading "Competing Interests Statement" with "Conflict of interest".

2) Figures and tables:

- Please make sure that all figures and figure panels are referenced in the main manuscript text. In particular, callouts for Fig 6E, Fig 8I/J (the figure is missing panel I), Fig EV2 panels are missing.
- The movies should be renamed Movie EV1/2. Both movies should be zipped with their legends.
- Table EV1 should be made Dataset EV1. The legend and file name need to be added in the dataset file, and the legend should be removed from the article file.
- Tables EV2 and EV3 should then become Table EV1 and EV2.
- There are 2 references to a Supplementary Table 1 which needs correcting/removing.

3) Checklist: Please provide relevant information in sections E/11-12.

4) Thank you for providing source data. Please upload them so as to have 1 pdf file per figure. Please also check the labeling of the source data, as uncropped blots labeled "Appendix figure 5D"

seem to match Appendix Figure 8B, and blots labeled "Appendix Figure 5B" seem to match Appendix Figure 8D.

5) Please provide 'The paper explained' section, which is a summary of the articles to emphasize the major findings in the paper and their medical implications for the non-specialist reader. Please provide a draft summary of your article highlighting

6) Thank you for providing a synopsis. I slightly edited it to match our style and format, please let me know if you agree with the following:

The disease mechanism linking the phospholamban (PLN) p.Arg14del mutation to dilated cardiomyopathy is incompletely understood. In this study, patient-derived human cardiomyocytes were used to elucidate this molecular mechanism.

- Sarcoplasmic reticulum function was unaltered in PLN p.Arg14del human cardiomyocytes.
- Impairment of Endoplasmic reticulum mitochondria interface was discovered as a novel disease phenotype.
- Cytoplasmic Ca²⁺ scavenging improved the disease phenotype and revealed the role for cytoplasmic Ca²⁺ in disease development.

Please also provide a visual abstract to illustrate your article as a png file 550 px-wide x 400-px high.

7) As part of the EMBO Publications transparent editorial process initiative (see our Editorial at <http://embomolmed.embopress.org/content/2/9/329>), EMBO Molecular Medicine will publish online a Review Process File (RPF) to accompany accepted manuscripts.

This file will be published in conjunction with your paper and will include the anonymous referee reports, your point-by-point response and all pertinent correspondence relating to the manuscript. Let us know whether you agree with the publication of the RPF and as here, **IF YOU WANT TO REMOVE OR NOT ANY FIGURES** from it prior to publication.

I look forward to receiving your revised manuscript.

Yours sincerely,

Lise Roth

Lise Roth, PhD
Editor
EMBO Molecular Medicine

***** Reviewer's comments *****

Referee #1 (Comments on Novelty/Model System for Author):

This manuscript defines a novel mechanism by which the PLN Arg14del mutation causes DCM, implicating abnormalities of the ER and mitochondria. In this well-designed study PLN Arg14del hiPSC-cardiomyocytes and isogenic control lines were generated by CRISPR/Cas9 technology and fully characterized in 3D engineered heart tissues. Complementary approaches including immunofluorescence, proteomics, RNA sequencing, electron microscopy and biochemical experiments demonstrate impaired ER/mitochondria structure in hiPSC-cardiomyocytes with the PLN Arg14del mutation. The study also includes rescue experiments which show improvement of the pathology by Ca²⁺ scavenging. The findings are clinically important because this PLN mutation causes cardiomyopathy in humans. An impressive amount of work was performed to establish a new mechanism underlying PLN Arg14del-mediated cardiomyopathy.

Referee #1 (Remarks for Author):

In this revised manuscript entitled "Impairment of the ER/mitochondria compartment in human cardiomyocytes with PLN p.Arg14del mutation" by Cuello et al., the authors performed additional experiments to address the major concerns identified in the first version. In particular:

- The issue with the immunofluorescence experiments (Figure 1e) has been addressed with new data showing reduced expression and, as expected, identical sub-cellular localization of PLN in PLN p.Arg14del hiPSC-CM and PLNic.
- The sample number for Western blot analyses, which was too low for many experiments, has been increased, which strengthens the data and the conclusion reached.
- Additional IF experiments were performed and show increased perinuclear aggregates in hiPSC-CM with the PLN Arg14Del mutation.
- The control empty lentivirus, that was lacking in the first version of the manuscript for the rescue experiment, is now provided.
- Results show improved ER/mitochondria, response to stress and contractile function after Ca²⁺ scavenging.

Overall, the additional experiments and new data generated significantly improve the paper.

Referee #2 (Comments on Novelty/Model System for Author)

No comments

Referee #2 (Remarks for Author):

This is a revised version of the manuscript. The authors have adequately responded to the different points raised by this reviewer. The paper has been substantially improved. No further comments. :

Referee #3 (Comments on Novelty/Model System for Author):

State-of-art techniques. The novelty is somehow undermined by the previous work in calcium in the IP-CMs with the same mutation, but this paper harbors elements of novelty. Medical impact, limited by the in vitro only work.

Referee #3 (Remarks for Author):

As results of the revision, the manuscript is significantly improved and elegantly addresses an important issue. No further comments.

The authors performed the requested changes.

7th Apr 2021

Dear Prof. Hansen,

Thank you for sending the revised files, and please accept my apologies for the delay in getting back to you.

I am pleased to say that your manuscript is now accepted for publication in EMBO Molecular Medicine!

Before we can transfer your manuscript to our publisher, please address the remaining editorial issues:

- Thank you for providing a table with exact n and p values. Please include the table in the Appendix (it should not be an EV table), and update the callout in the text accordingly.
- Please remove/update the references to Supplementary Table 1 on p. 11 line 263 and p. 36 line 915.
- Thank you for depositing your datasets in public repositories. Please note that these datasets have to be made public before publication of your manuscript.
- Thank you for providing a nice synopsis picture. Could you please send us a jpeg or png version of the figure?

Please use the attached document for any further modification. Once revised, you may send us the files via email.

Thank you very much for bearing with these last editorial matters.

Congratulations on a nice study!

With my best wishes,

Lise Roth

Lise Roth, PhD
Editor
EMBO Molecular Medicine

Corresponding Author Name: Arne Hansen

Journal Submitted to: EMBO Mol Med

Manuscript Number: EMM-2020-13074